# Prostaglandin in the ventromedial hypothalamus regulates peripheral glucose metabolism

Ming-Liang Lee [1], Hirokazu Matsunaga[1], Yuki Sugiura[2], Takahiro Hayasaka[3], Izumi Yamamoto[1], Taiga Ishimoto[1], Daigo Imoto[1], Makoto Suematsu [2], Norifumi Iijima[4,5], Kazuhiro Kimura [1], Sabrina Diano [6,7] & Chitoku Toda [1✉]

The hypothalamus plays a central role in monitoring and regulating systemic glucose metabolism. The brain is enriched with phospholipids containing poly-unsaturated fatty acids, which are biologically active in physiological regulation. Here, we show that intraperitoneal glucose injection induces changes in hypothalamic distribution and amounts of phospholipids, especially arachidonic-acid-containing phospholipids, that are then metabolized to produce prostaglandins. Knockdown of cytosolic phospholipase A2 (cPLA2), a key enzyme for generating arachidonic acid from phospholipids, in the hypothalamic ventromedial nucleus (VMH), lowers insulin sensitivity in muscles during regular chow diet (RCD) feeding. Conversely, the down-regulation of glucose metabolism by high fat diet (HFD) feeding is improved by knockdown of cPLA2 in the VMH through changing hepatic insulin sensitivity and hypothalamic inflammation. Our data suggest that cPLA2-mediated hypothalamic phospholipid metabolism is critical for controlling systemic glucose metabolism during RCD, while continuous activation of the same pathway to produce prostaglandins during HFD deteriorates glucose metabolism.

[1] Laboratory of Biochemistry, Graduate School of Veterinary Medicine, Hokkaido University, Sapporo, Hokkaido, Japan. [2] Department of Biochemistry, Keio University School of Medicine, Shinjuku-ku, Tokyo, Japan. [3] Department of Gastroenterological Surgery I, Graduate School of Medicine, Hokkaido University, Sapporo, Hokkaido, Japan. [4] National Institutes of Biomedical Innovation, Health and Nutrition, Ibaraki, Osaka, Japan. [5] Immunology Frontier Research Center, Osaka University, Suita, Osaka, Japan. [6] Department of Molecular Pharmacology and Therapeutics, Columbia University Irving Medical Center, New York, USA. [7] Department of Cellular and Molecular Physiology, Yale School of Medicine, Yale University, New Haven, CT, USA. ✉email: c-toda@vetmed.hokudai.ac.jp

Recent evidence from animal models indicates that the brain plays a critical role in the systemic regulation of glucose metabolism[1,2]. Neurons in the hypothalamus integrate hormonal and nutritional information and maintain glucose homeostasis by controlling metabolism in peripheral tissues. Numerous brain regions have been reported to maintain whole-body glucose homeostasis[3–5]. In particular, the ventromedial nucleus of the hypothalamus (VMH) and arcuate nucleus of the hypothalamus (ARC) are critical nuclei for the glucose metabolism. Obesity can attenuate the function of these nuclei and promotes type II diabetes via hypothalamic inflammation[6]. However, the hypothalamic mechanism that regulates systemic glucose metabolism is not fully understood.

The VMH has important roles in regulating glucose metabolism in peripheral tissues[7], and the majority of neurons in the VMH express steroidogenic factor 1 (Sf1). Photoactivation of Sf1 neurons increases hepatic glucose production (HGP)[4,8] and simultaneously enhances insulin sensitivities in the liver, muscle and brown adipose tissue (BAT)[9]. Leptin regulates the neuronal activity of Sf1 neurons, and increases glucose utilization and insulin sensitivity in peripheral tissues[10–12]. There are two main neuronal populations in the ARC, the orexigenic agouti-related peptide (AgRP) neurons, which co-express neuropeptide Y (NPY), and the anorexigenic proopiomelanocortin (POMC) neurons[13]. POMC and AgRP neurons control HGP in opposite ways, i.e., activation of POMC increases insulin sensitivity in the liver, while AgRP activation decreases liver insulin sensitivity[14,15]. Hormones, including insulin, leptin, and ghrelin, regulate glucose metabolism by changing the activities of these neurons and their gene expression. In addition, subpopulations of Sf1, POMC, and AgRP neurons are also activated (glucose excited neurons) or inhibited (glucose inhibited neurons) by glucose[16].

Fatty acids regulate the activities of hypothalamic neurons[16] and the lipid metabolism within the hypothalamus plays important roles in energy balance and glucose metabolism[17,18]. Phospholipids with biologically active polyunsaturated fatty acids (PUFAs), including phosphatidyl-inositol (PI), phosphatidyl-ethanolamine (PE), and phosphatidyl-serine (PS), are abundantly found in the brain[19]. Some membrane phospholipids generate free PUFAs to regulate physiological functions of the brain. For example, phospholipase A2 (PLA2) preferentially generates arachidonic acid (AA) from phospholipids[20]. AA plays roles in several physiological functions, including thermogenesis in BAT and increasing blood glucose levels[21]. AA is also the precursor for eicosanoids such as, prostaglandin and hydroxyeicosatetraenoic acid (HETE). Other PUFAs, such as oleic acid (OA), modulate activities of nutrient-sensing neurons to regulate insulin secretion[22], and intracerebroventricular injection of OA enhances insulin sensitivity in the liver[23]. However, the distributions of FAs, PUFAs, and PUFA-containing phospholipids in the hypothalamus and their roles in whole-body glucose metabolism are not clearly understood.

Here, we explore hypothalamic lipid metabolism in the regulation of systemic glucose homeostasis and its potential role in the development of diabetes using imaging mass spectrometry (IMS). We found that glucose injection in mice fed on a regular chow diet induces a decrease in phospholipids containing AA, which is mediated by cytosolic phospholipase A2 (cPLA2). Prostaglandins produced from phospholipids in the hypothalamus activates VMH neurons and increases insulin sensitivity in skeletal muscles. However, hypothalamic cPLA2-mediated prostaglandin production is enhanced by high-fat diet and induces neuroinflammation, and blockage of this enzyme confers resistance to developing diabetes.

## Results

**Hyperglycemia decreases phospholipids and produces prostaglandins in the hypothalamus.** To determine the distribution of FAs and phospholipids, hypothalamic slices were examined by IMS (Fig. 1). The signal intensities of palmitic acid (PA), stearic acid (SA), AA, and docosahexaenoic acid (DHA) were high around the third ventricle and the ventrolateral region of the hypothalamus in C57BL/6J mice (Fig. 1a). Similar distributions of phospholipids, such as phosphatidyl-ethanolamine (PE; 18:0/20:4), phosphatidyl-inositol (PI; 18:0/20:4), and PI (18:1/20:4) were also observed. However, signal intensities for phosphatidyl-serine (PS; 18:0/20:4) were low around the third ventricle while PS (18:0/22:6) distribution was ubiquitously observed (Fig. 1b). We then measured hypothalamic lipids in mice that received an intraperitoneal (i.p.) glucose injection. Signal intensities for PI (18:0/20:4), PI (18:1/20:4) and PE (18:0/20:4) were significantly decreased in the VMH after glucose administration (Fig. 1c, d). Similarly, the signal intensities for PI (18:0/20:4), PI (18:1/20:4), PE (18:0/20:4), and PS (18:0/16:0) were decreased in the ARC after glucose injection (Fig. 1c,e). Hydrolysis of these phospholipids generates FAs, including AA (20:4), oleic acid (OA; 18:1), PA (16:0), or SA (18:0). However, the signal intensities of the four fatty acids were not changed after glucose injection, neither in the VMH nor ARC (Fig. 1f-h). Hydrolysis of these phospholipids also generates lysospecies, such as lysophosphatidyl-inositol, -ethanolamine, and -serine. Signal intensities for lysospecies in the VMH and ARC tended to be increased after glucose injection (Supplementary Fig. 1).

AA is the source of eicosanoids and AA metabolism is catalyzed by enzymes such as cyclooxygenase (COX), lipoxygenase, and cytochrome P450. To elucidate if glucose injection increased AA metabolism, we examined the effect of glucose injection on eicosanoid production in the whole hypothalamus using a liquid chromatography–mass spectrometry (LC-MS). Compared with saline, glucose injection increased COX-mediated hypothalamic production of prostaglandins, including 6-keto-PGF1α, PGD2, 13,14-dihydro-15-keto-PGF2α and PGE2 (Fig. 1i–m). Lipoxygenase-mediated production of 12-HETE was increased in glucose-injected mice (Supplementary Fig. 2a). However, most of the lipoxygenase- and cytochrome P450-mediated production of HETEs and EETs was not detected or changed by glucose injection (Supplementary Fig. 2). Thus, the data suggests that increased glucose levels decreases AA-containing phospholipids to produce prostaglandins.

**Blocking the PLA2-mediated pathway in the hypothalamus impairs systemic glucose metabolism.** PLA2 is the primary enzyme that generates AA from phospholipids[24]. We then investigated the role of PLA2-mediated phospholipid utilization in glucose metabolism during acute hyperglycemia after an intrahypothalamic administration of methyl arachidonyl fluorophosphonate (MAFP), a PLA2 inhibitor. MAFP-injected mice showed decreased glucose tolerance compared to vehicle-injected mice and no changes in circulating insulin levels were observed (Fig. 2a,b). Hypothalamic injection of indomethacin, an inhibitor of COX1/2, also impaired glucose tolerance (Fig. 2c,d) and increased blood glucose levels after refeeding (Fig. 2e), suggesting that prostaglandins regulate hypothalamic function to decrease blood glucose levels. However, intra-hypothalamic injection of phospholipase C (PLC) inhibitor, U73122, or IP3 receptor antagonist, xestospondin 2, did not affect glucose tolerance (Supplementary Fig. 3). Thus, our results suggest that the PLA2-mediated AA release and production of prostaglandins by COX1/2 in the hypothalamus, but not PLC-IP3 pathway, play a role in regulating glucose metabolism during acute hyperglycemia.

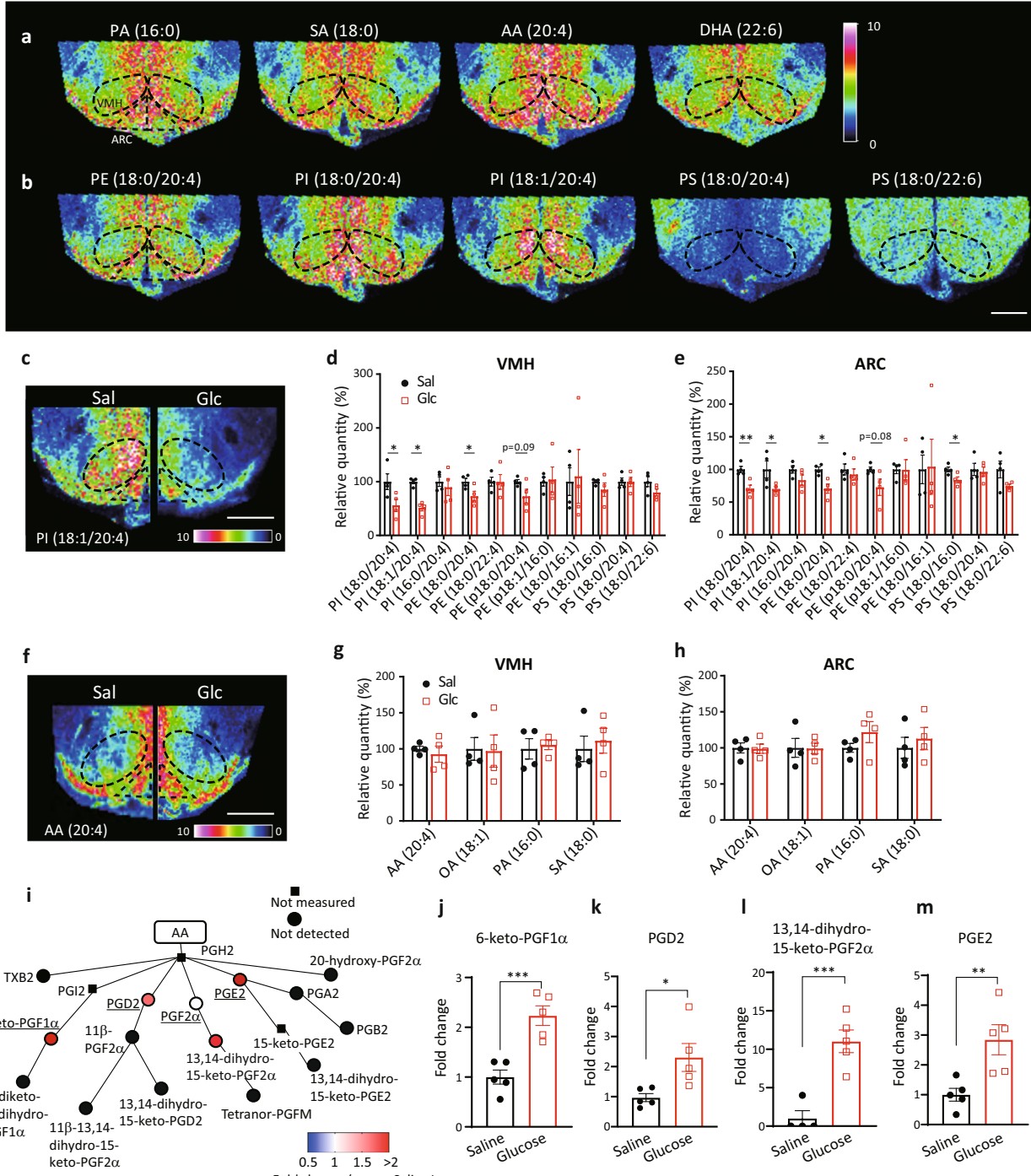

**Fig. 1 Hyperglycemia increases prostaglandin production derived from phospholipids. a, b** Representative results of imaging mass spectrometry (IMS) showing distributions of hypothalamic fatty acids (**a**) and phospholipids (**b**) from untreated RCD-fed mice. The dashed black line shows the position of the VMH. Scale bar: 500 μm. **c–h** Distributions of phospholipids and fatty acids in the hypothalamus 30 min after injection of saline (Sal) or glucose (Glc; 2 g/ kg). **c, f** Representative results of IMS on hypothalamic phosphatidyl-inositol (PI; 18:1/20:4) (**c**) and arachidonic acid (AA) (**f**) from mice i.p. injected with saline (left half) or glucose (right half). Scale bar: 500 μm. **d, e** Relative intensities of phospholipids in the VMH (**d**) and ARC (**e**) after injection with saline (n = 4) or glucose (n = 4). (two-tailed t test for each molecule, VMH: p = 0.0268 in PI (18:0/20:4), p = 0.0005 in PI (18:1/20:4), and p = 0.0491 in PE (18:0/20:4); ARC: p = 0.0073 in PI (18:0/20:4), p = 0.0347 in PI (18:1/20:4), p = 0.0106 in PE (18:0/20:4), and p = 0.0331 in PS(18:0/16:0), Glc vs Sal **g**, **h** Relative intensities of fatty acids in the VMH (**g**) and ARC (**h**) after injection with saline (n = 4) or glucose (n = 4). **i–m** LC-MS results showing the effects of glucose injection on AA metabolites in the whole hypothalamus. **i** Relative amounts of hypothalamic prostaglandins mediated by cyclooxygenase. Major prostaglandins were underlined. **j** 6-keto-PGF1α, **k** PGD2, **l** 13,14-dihydro-15-keto-PGF2α and **m** PGE2 were increased by glucose injection (two-tailed t test, p = 0.0009 in **j**, p = 0.0244 in **k**, p = 0.0011 in **l**, p = 0.0099 in **m**, n = 5/each, Glc vs Sal). **d–h** and **j–m** represent the mean ± SEM; *p < 0.05; **p < 0.01; ***p < 0.001. **i** represents the mean fold change in color. PA palmitic acid, SA stearic acid, AA arachidonic acid, DHA docosahexaenoic acid, PE phosphatidyl-ethanolamine, PI phosphatidyl-inositol, PS phosphatidyl-serine.

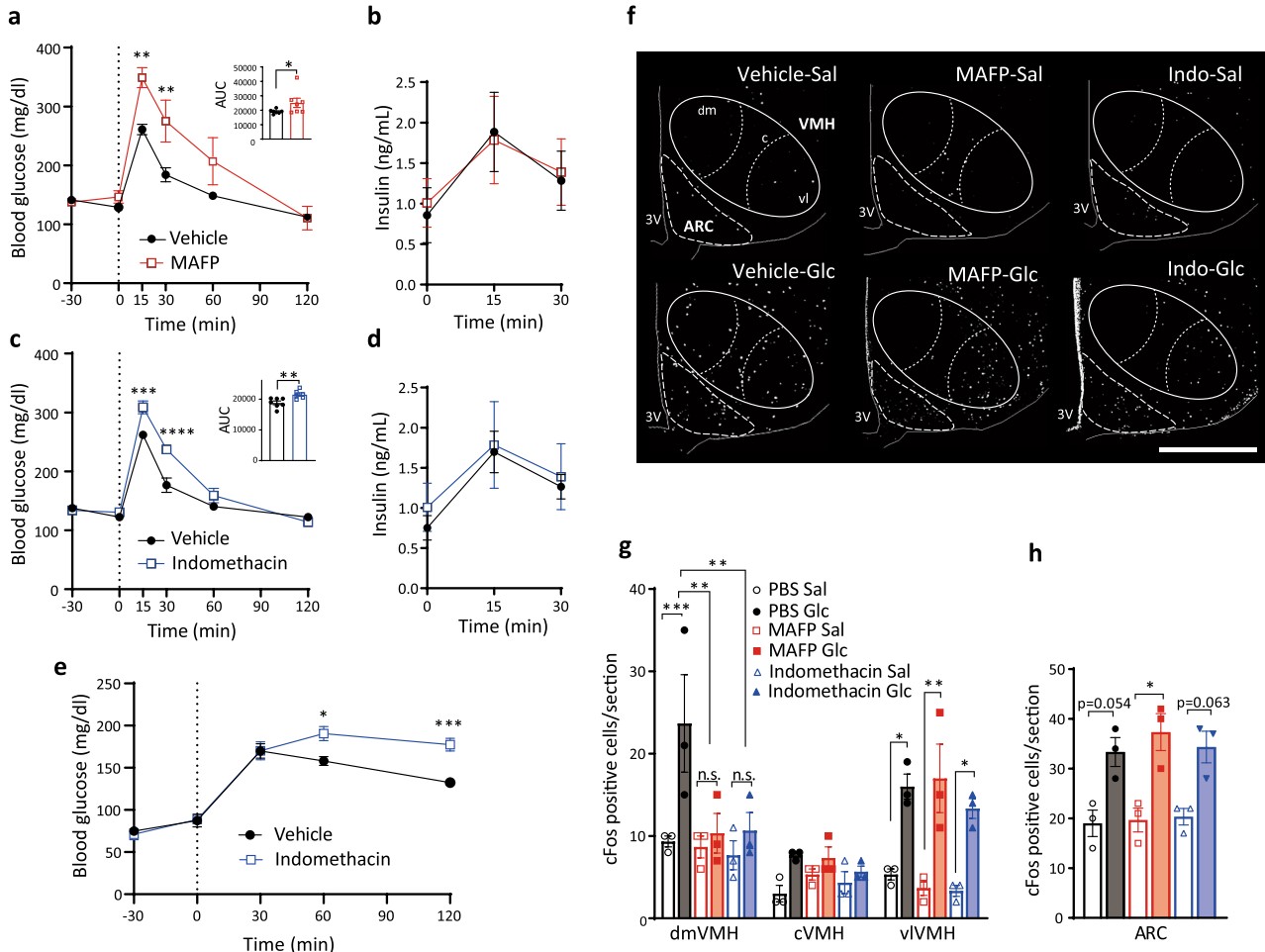

**Fig. 2 Hypothalamic PLA2- and COX-mediated AA metabolism regulates systemic glucose tolerance and modulates glucose responsiveness in the dmVMH. a** Glucose tolerance test (GTT; 0–120 min) after intra-hypothalamic injection (−30 min) of MAFP ($n = 7$) or vehicle ($n = 7$; two-way ANOVA followed by Sidak multiple comparison test, $p = 0.0065$ at time = 15, $p = 0.0045$ at time = 30, MAFP vs Vehicle; two-tailed $t$ test in area under the curve (AUC) during GTT, $p = 0.0422$, MAFP vs Vehicle). **b** Blood insulin concentration of MAFP ($n = 6$) or vehicle ($n = 6$) injected mice during GTT. **c** GTT (0–120 min) after intra-hypothalamic injection (−30 min) of indomethacin ($n = 6$) or vehicle ($n = 7$; two-way ANOVA followed by Sidak multiple comparison test, $p = 0.0002$ at time = 15, $p < 0.0001$ at time = 30, indomethacin vs Vehicle; two-tailed $t$ test in AUC during GTT, $p = 0.0074$, indomethacin vs Vehicle). **d** Blood insulin concentration of indomethacin ($n = 6$) or saline ($n = 7$) injected mice during GTT. **e** Blood glucose levels in refeeding after intra-hypothalamic injection (−30 min) of indomethacin ($n = 6$) or vehicle ($n = 5$; two-way ANOVA followed by Sidak multiple comparison test, $p = 0.0141$ at time = 60, $p = 0.0004$ at time = 120, indomethacin vs Vehicle). **f** Representative micrographs showing immunofluorescent cFos staining in the hypothalamus of saline (upper panels) or glucose (lower panels) injected mice after i.c.v. injection of PBS, MAFP, or indomethacin (indo). Scale bar: 500 μm. dm dorsomedial, c central, vl ventrolateral part of the VMH. **g, h** Quantification of cFos expression in the dorsomedial (dmVMH), central (cVMH), and ventrolateral (vlVMH) subregions of the VMH (**g**) and ARC (**h**) from mice injected with saline or glucose after i.c.v. injection of PBS, MAFP, or indomethacin ($n = 3$ in each experimental group; VMH: two-way ANOVA followed by Sidak multiple comparison test, for dmVMH: $p = 0.0004$ PBS Glc vs PBS Sal, $p = 0.0011$ PBS Glc vs MAFP Glc, $p = 0.0015$ PBS Glc vs Indomethacin Glc, for vlVMH, $p = 0.0145$ PBS Sal vs PBS Glc, $p = 0.0011$ MAFP Sal vs MAFP Glc, $p = 0.0269$ Indomethacin Sal vs Indomethacin Glc. For ARC: one-way ANOVA followed by Sidak multiple comparison test, $p = 0.0124$ MAFP Sal vs MAFP Glc. All data represent the mean ± SEM; *$p < 0.05$; **$p < 0.01$; ***$p < 0.001$; ****$p < 0.0001$.

**PLA2-mediated production of prostaglandin is necessary for the responsiveness of the VMH to glucose.** To understand the role of PLA2 in controlling glucose metabolism, we examined the effect of PLA2 inhibitors on hypothalamic neuronal activation by cFos expression. Vehicle, MAFP, or indomethacin were injected intracerebroventricularly (i.c.v.) 30 min prior to i.p. injection of either saline or glucose in fasted mice. In i.c.v. vehicle-injected mice, glucose administration increased cFos-positive cells in the dorsomedial (dm) and ventrolateral (vl) regions of the VMH and in the ARC (Fig. 2f–h). In i.c.v. MAFP-injected mice, glucose did not alter the number of cFos-positive neurons in the dmVMH (Fig. 2g). An increase in cFos-positive neurons after glucose injection was still detected in the vlVMH and ARC compared

with saline injected mice (Fig. 2g, h). Similar results were observed in i.c.v. indomethacin-injected mice after an i.p. injection of glucose (Fig. 2g, h). Taken together, these data showing that both MAFP and indomethacin block neuronal activation during acute hyperglycemia in the dmVMH, indicates that metabolites of phospholipid-derived prostaglandins regulate glucose responsiveness of neurons in the dmVMH, while glucose activates neurons in the vlVMH and ARC independently of PLA2 and COX1/2.

The role of PLA2 in the leptin-induced neuronal activation was also assessed in the i.c.v. MAFP-injected mice. In i.c.v. vehicle-injected mice, leptin injection increased cFos-positive cells in the dmVMH (Supplementary Fig. 4a–c) and it also increased

pSTAT3-positive cells in the dmVMH and ARC (Supplementary Fig. 4d–f). The leptin-induced increases in cFos-positive cells were decreased by i.c.v. injection of MAFP (Supplementary Fig. 4). These results suggest that cPLA2 also regulates neuronal activity in the dmVMH in response to peripheral hormones, such as leptin.

**Knockdown of *pla2g4a* in Sf1 neurons impairs glucose metabolism in regular chow diet feeding.** Next, to explore the role of PLA2 in VMH neurons, short hairpin RNA (shRNA) against *pla2g4a*, a gene encoding cytosolic PLA2 (cPLA2), which has specificity for *sn*-2 arachidonic acid and a role in eicosanoid production[20], was transfected to the VMH through an adeno-associated virus (AAV) cre-recombinase (cre)-dependent in Sf1-cre mice (Supplementary Fig. 5a, b). Expression of *pla2g4a* mRNA was significantly decreased in the VMH of Sf1-cre mice injected with AAV-DIO-shRNA (cPLA2KD$^{Sf1}$) compared with AAV-DIO-GFP (GFP$^{Sf1}$)-injected mice (Supplementary Fig. 5c). The hypothalamic cPLA2 activity was not changed by glucose injection in C57BL/6J mice (Supplementary Fig. 5d). The activity was measured by the tissue homogenate of the hypothalamus, in which the cellular calcium concentration was not conserved. Thus, the hypothalamic cPLA2 activity in response to glucose may be regulated by cytosolic calcium concentration[24]. However, the decrease of AA-contained-phospholipids in the VMH by glucose injection was blocked by shRNA transfection in cPLA2KD$^{Sf1}$ mice (Supplementary Fig. 5e, f). In the ARC, these phospholipids were still decreased by glucose (Supplementary Fig. 5e, g), suggesting the AAV-induced shRNA expression only affects neurons in the VMH rather than in the ARC. We further confirmed if the cre-dependent shRNA expression is restricted in the VMH without leaking to the ARC. The AAV-DIO-GFP was injected into the VMH of Sf1-cre mice and α-melanocyte stimulating hormone (α-MSH) was stained as an indicator of POMC neurons. In the image (Supplementary Fig. 5h), POMC neurons appear to be distributed in the main part of the ARC, but fewer in the dorsal part. GFP-positive neurons were not merged with POMC neurons (Supplementary Fig. 5h, i), suggesting that the AAV-mediated gene expression occurred in the VMH, but not in the ARC.

After 8 weeks of viral injection, knockdown of cPLA2 in the Sf1 neurons decreased glucose tolerance and insulin sensitivity compared to GFP$^{Sf1}$ mice (Fig. 3a, b), although it did not change body weight or weights of adipose tissues, muscle, and liver (Supplementary Fig. 6a). These changes in the glucose metabolism depend on the accuracy of the AAV injection, since the mice with missed injection site of the virus had no effects on glucose and insulin tolerance (Supplementary Fig. 6b–d). cPLA2KD$^{Sf1}$ mice also increased blood glucose levels after refeeding compared with GFP$^{Sf1}$ mice (Fig. 3c). 2-deoxy-glucose (2DG)-induced hyperglycemia, which represents a glucose deprivation-induced counter-regulatory responses, was comparable between groups (Fig. 3d). To rule out the involvement of astrocytic cPLA2, AAV-GFAP-Cre, and AAV-DIO-shRNA against *pla2g4a* were co-injected into the hypothalamus of wild type mice to knockdown the expression of *pla2g4a* in hypothalamic astrocytes (Supplementary Fig. 7a, b). The knockdown of cPLA2 in astrocytes did not alter glucose metabolism, insulin sensitivity, or body weight compared with control mice (Supplementary Fig. 7c–e). Thus, our data suggest that cPLA2 in Sf1 neurons, not astrocytes, regulates peripheral glucose metabolism.

To further investigate the role of cPLA2 in Sf1 neurons in glucose metabolism, we next performed hyperinsulinemic–euglycemic clamp studies. cPLA2KD$^{Sf1}$ mice showed a lower glucose infusion rate (GIR) to maintain euglycemia compared with GFP$^{Sf1}$ mice

(Fig. 3e–g). The rate of disappearance (Rd) and glycolysis were also lower in cPLA2KD$^{Sf1}$ mice compared with GFP$^{Sf1}$ mice (Fig. 3h, i). However, endogenous glucose production (EGP) was not different between the two groups (Fig. 3j, k), suggesting that glucose utilization, rather than EGP, was impaired in cPLA2KD$^{Sf1}$ mice. In agreement with this, cPLA2KD$^{Sf1}$ mice showed decreased 2DG uptake in the red part of gastrocnemius muscle (GR) compared with control mice (Fig. 3l). 2DG uptake in white adipose tissue (WAT) and the brain (cortex) were similar between groups (Fig. 3m, n).

To assess changes in neuronal activation, we next analyzed cFos expression in cPLA2KD$^{Sf1}$ mice compared with controls. Glucose-induced cFos expression in the dmVMH of cPLA2KD$^{Sf1}$ mice was blunted compared with glucose-injected control mice (Fig. 3o, p). The glucose-induced cFos expression in either vlVMH or ARC was not changed after the knockdown of cPLA2 (Fig. 3p, q).

Taken together, our data suggest that cPLA2-mediated prostaglandin production regulates glucose-induced activation of dmVMH neurons to control insulin sensitivity in muscle.

**High-fat diet decreases AA-containing phospholipids and produces prostaglandins in the hypothalamus.** High-fat diet (HFD) induces inflammation and impairs hypothalamic functions[25]. Long chain fatty acyl CoA, a proinflammatory signal, accumulates in the hypothalamus during HFD feeding[26]. Thus, we examined the effect of HFD on lipid distribution in the hypothalamus. In mice fed an HFD for 8 weeks, the signal intensities for FAs, including AA, were greater in the ARC but not the VMH than those observed in control mice fed a RCD (Fig. 4a–c). However, signal intensities for phospholipids in the hypothalamus were lower in HFD-fed mice (Fig. 4d–f). In both VMH and ARC, the signal intensities for PI (18:0/20:4), PI (18:1/ 20:4), PE (18:0/20:4), PE (p18:0/20:4), and PS (18:0/22:6) were significantly decreased in HFD-fed mice (Fig. 4e, f). The signal intensity for lysophosphatidyl-inositol (LPI; 18:0) was also higher in the VMH and ARC of HFD-fed mice compared to RCD-fed mice (Supplementary Fig. 8). Because PLA2 generates AA from these phospholipids to regulate cellular activities, we next analyzed the activity of hypothalamic cPLA2 and found that cPLA2 activity was higher in HFD-fed mice compared with RCD-fed mice (Fig. 4g). However, the activity of secretory-phospholipase A2 (sPLA2) remained similar between RCD- and HFD-fed mice (Fig. 4h). The phosphorylation of cPLA2 at Ser505, which stimulates its enzymatic activity[27], was measured by western blotting (Fig. 4i, j). The phosphorylation of cPLA2 in the hypothalamus was higher in the HFD-fed mice compared with RCD-fed mice, while total amount of cPLA2 was not changed (Fig. 4i, k).

We next explored the effect of HFD on the production of eicosanoids in the hypothalamus by LC-MS (Fig. 4l and Supplementary Fig. 9). In HFD-fed mice, AA tended to be higher but did not reach significant difference (Fig. 4m). COX-mediated production of prostaglandins, including PGD2, PGF2α, PGE2, 11-beta-13,14-dihydro-15-keto-PGF2α, 13,14-dihydro-15-keto-PGD2, and 20-hydroxy-PGF2α, was increased in HFD-fed mice compared to RCD-fed mice (Fig. 4n–s). Among lipoxygenase-mediated eicosanoids, only 12-HETE was significantly increased after HFD feeding (Supplementary Fig. 9).

**Knockdown of *pla2g4a* improves HFD-induced impairment of glucose metabolism and recovers glucose responsiveness of the vlVMH and ARC to hyperglycemia.** To understand the role of HFD-induced activation of hypothalamic cPLA2, we fed cPLA2KD$^{Sf1}$ mice with HFD for 8 weeks and examined the role

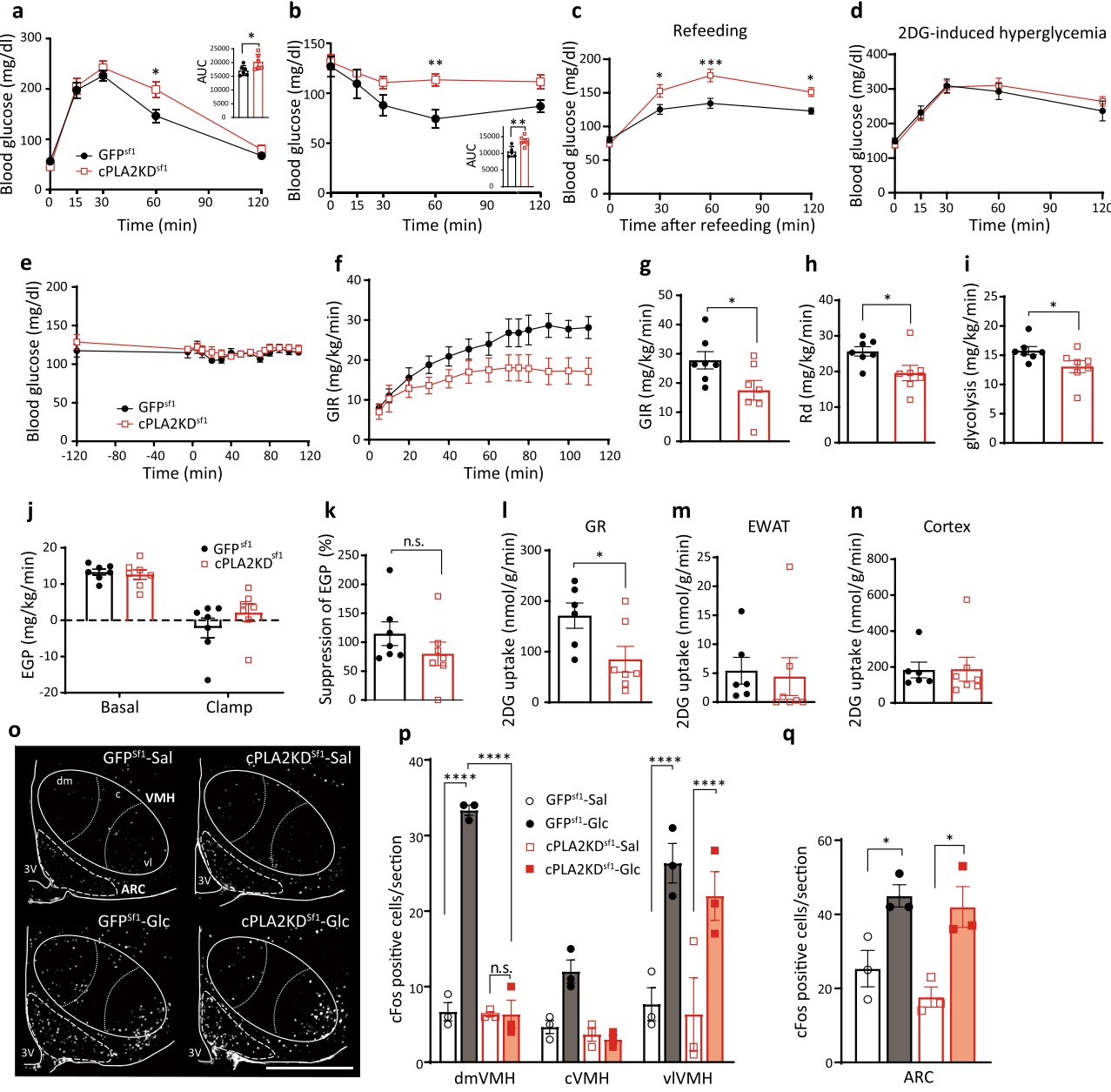

of cPLA2 on glucose metabolism. Body weight and tissue weight of HFD-fed cPLA2KD[Sf1] mice (cPLA2KD[Sf1]-HFD) were comparable to those of HFD-fed control GFP[Sf1] mice (GFP[Sf1]-HFD; Fig. 5a and Supplementary Fig. 10). Unlike RCD-fed mice (Fig. 3a), knockdown of cPLA2 in Sf1 neurons increased glucose tolerance (Fig. 5b). However, insulin tolerance test showed no difference between groups (Fig. 5c). The similar results were also observed in female mice. The body weight was not changed between female GFP[Sf1]-HFD and female cPLA2KD[Sf1]-HFD mice (Supplementary Fig. 11a). Knockdown of cPLA2 in Sf1 neurons increased glucose tolerance but not insulin sensitivity (Supplementary Fig. 11b, c). The tissue weight was also not changed by knockdown of cPLA2 in Sf1 neurons of female mice (Supplementary Fig. 11d).

In GFP[Sf1]-HFD mice, no significant changes in cFos-positive neurons in the VMH or ARC were observed after glucose injection compared with saline injection (Fig. 5d–f), suggesting that HFD inhibits glucose sensing in the VMH and the ARC, which has been already reported in previous studies[28–30].

However, in cPLA2KD[Sf1]-HFD, the number of cFos-positive neurons were significantly higher in the vlVMH and ARC after glucose injection compared with saline injection, suggesting that cPLA2 knockdown improved neuronal responsiveness to glucose under the HFD-fed condition (Fig. 5d–f).

To understand the role of hypothalamic cPLA2 on glucose metabolism in HFD-fed mice, we performed hyperinsulinemic–euglycemic clamping (Fig. 5g–m). To maintain euglycemia, lower GIR was required in GFP[Sf1]-HFD (Fig. 5i, black) than RCD-fed GFP[Sf1] mice (Fig. 3g, black), suggesting that the HFD-fed mice became insulin resistant and thus lower glucose was needed to maintain euglycemia in the HFD-fed mice than RCD-fed mice. In the HFD-fed groups, significantly higher GIR was required in cPLA2KD[Sf1]-HFD than in GFP[Sf1]-HFD (Fig. 5g–i). Unlike RCD-fed mice (Fig. 3h, i), glucose utilization (Rd and glycolysis) in HFD-fed mice was not altered by knocking down cPLA2 (Fig. 5j, k). In contrast, insulin inhibition of EGP was stronger during the clamp period in cPLA2KD[Sf1]-HFD mice than in GFP[Sf1]-HFD (Fig. 5l, m). These results suggest that cPLA2 in the VMH

**Fig. 3 Knockdown of Sf1-neuronal *pla2g4a* impairs systemic glucose metabolism. a** Glucose tolerance test in cPLA2KD$^{Sf1}$ ($n = 6$) and GFP$^{Sf1}$ mice ($n = 6$; two-way ANOVA followed by Sidak multiple comparison test, $p = 0.0126$ at time = 60, cPLA2KD$^{Sf1}$ vs GFP$^{sf1}$; two-tailed $t$ test in area under the curve (AUC) during GTT, $p = 0.0433$, cPLA2KD$^{Sf1}$ vs GFP$^{sf1}$). **b** Insulin tolerance test in cPLA2KD$^{Sf1}$ ($n = 6$) and GFP$^{Sf1}$ mice ($n = 5$; two-way ANOVA followed by Sidak multiple comparison test, $p = 0.0094$ at time = 60 cPLA2KD$^{Sf1}$ vs GFP$^{sf1}$; two-tailed $t$ test in AUC during GTT, $p = 0.0071$, cPLA2KD$^{Sf1}$ vs GFP$^{sf1}$). **c** Blood glucose levels after refeeding in cPLA2KD$^{Sf1}$ ($n = 6$) and GFP$^{Sf1}$ mice ($n = 9$; two-way ANOVA followed by Sidak multiple comparison test, $p = 0.0255$ at time = 30, $p = 0.0003$ at time = 60, $p = 0.0211$ at time = 120, cPLA2KD$^{Sf1}$ vs GFP$^{sf1}$). **d** 2-deoxy-glucose (2DG)-induced hyperglycemia in cPLA2KD$^{Sf1}$ ($n = 6$) and GFP$^{Sf1}$ mice ($n = 6$). **e–n** Hyperinsulinemic–euglycemic clamp studies in cPLA2KD$^{Sf1}$ and GFP$^{Sf1}$ mice. **e** Blood glucose levels during hyperinsulinemic–euglycemic clamp studies in cPLA2KD$^{Sf1}$ or GFP$^{Sf1}$ mice. **f** The glucose infusion rate (GIR) required to maintain euglycemia during the clamp period in cPLA2KD$^{Sf1}$ ($n = 7$) or GFP$^{Sf1}$ mice ($n = 7$). **g** The average GIR between 75 and 115 min in cPLA2KD$^{Sf1}$ ($n = 7$) or GFP$^{Sf1}$ mice ($n = 7$; two-tailed $t$ test, $p = 0.0395$, cPLA2KD$^{Sf1}$ vs GFP$^{sf1}$). **h** The rate of glucose disappearance (Rd) during the clamp period, which represents whole-body glucose utilization (two-tailed $t$ test, $p = 0.0355$, cPLA2KD$^{Sf1}$ vs GFP$^{sf1}$). **i** The rates of whole-body glycolysis in cPLA2KD$^{Sf1}$ ($n = 7$) or GFP$^{Sf1}$ mice ($n = 7$; two-tailed $t$ test, $p = 0.0497$, cPLA2KD$^{Sf1}$ vs GFP$^{sf1}$). **j** Endogenous glucose production (EGP) during both the basal and clamp periods in cPLA2KD$^{Sf1}$ ($n = 7$) or GFP$^{Sf1}$ mice ($n = 7$). **k** Insulin-induced suppression of EGP, which represents hepatic insulin sensitivity in cPLA2KD$^{Sf1}$ ($n = 7$) or GFP$^{Sf1}$ mice ($n = 7$). **l–n** Graphs showing 2-[$^{14}$C]-Deoxy-D-Glucose uptake in red portions of the gastrocnemius (GR; **l**), white adipocyte (EWAT; **m**) and brain (cortex; **n**) during the clamp period in cPLA2KD$^{Sf1}$ ($n = 7$) or GFP$^{Sf1}$ mice ($n = 7$; two-tailed $t$ test, $p = 0.0352$ in **l**, cPLA2KD$^{Sf1}$ vs GFP$^{sf1}$). **o** Representative micrographs showing immunofluorescent cFos staining in the hypothalamus of cPLA2KD$^{Sf1}$ and GFP$^{Sf1}$ mice after saline or glucose injection (3 g/kg). Scale bar: 500 μm. **p**, **q** Quantification of cFos expression in the dmVMH, cVMH, vlVMH, and ARC of cPLA2KD$^{Sf1}$ or GFP$^{Sf1}$ mice after saline ($n = 3$) or glucose ($n = 3$) injection (3 g/kg; VMH: two-way ANOVA followed by Sidak multiple comparison test, for dmVMH: $p < 0.0001$ GFP$^{Sf1}$ Sal vs GFP$^{Sf1}$ Glc, $p < 0.0001$ GFP$^{Sf1}$ Glc vs cPLA2KD$^{Sf1}$ Glc, for vlVMH: $p < 0.0001$ GFP$^{Sf1}$ Sal vs GFP$^{Sf1}$ Glc, $p < 0.0001$ cPLA2KD$^{Sf1}$ Sal vs cPLA2KD$^{Sf1}$ Glc. For ARC: one-way ANOVA followed by Sidak multiple comparison test, $p = 0.0425$ GFP$^{Sf1}$ Sal vs GFP$^{Sf1}$ Glc, $p = 0.0174$ cPLA2KD$^{Sf1}$ Sal vs cPLA2KD$^{Sf1}$ Glc.) All data represent the mean ± SEM; *$p < 0.05$; **$p < 0.01$; ***$p < 0.001$; ****$p < 0.0001$.

attenuates hepatic insulin sensitivity during HFD feeding. These data suggest that the role of cPLA2 and the mechanism to change glucose metabolism and neuronal activity by prostaglandins are different between HFD and RCD.

**Knockdown of cPLA2 in Sf1 neurons prevents hypothalamic inflammation.** We examined the effect of cPLA2 knockdown on hypothalamic inflammation, which attenuates neuronal functions, in HFD-fed mice. The mice were fed with a RCD or HFD for 8 weeks, and inflammation was measured by comparing the number and cell size of astrocytes and microglia. For this experiment, we used GFAP and Iba1 as markers of astrocytes and microglia, respectively (Fig. 6).

The number of GFAP-positive cells was increased in the ARC but not in the VMH after the control mice were fed with an HFD (Fig. 6a–e). The size of GFAP cells was enlarged in the ARC, but not in the VMH after mice were fed with an HFD (Fig. 6f, g). Compared to GFP$^{Sf1}$-HFD mice, cPLA2KD$^{Sf1}$-HFD had a decreased number of GFAP-positive cell and a smaller size of GFAP-positive cells in the ARC (Fig. 6e, g). No difference in GFAP cell number and size were found in the VMH between groups (Fig. 6d, f).

HFD also increased the number of Iba1-positive cells in the ARC, but not in the VMH (Fig. 6h–l). However, the somatic size of Iba1 cells was significantly larger in both VMH and ARC after mice were fed with an HFD compared to RCD (Fig. 6m, n), suggesting the HFD feeding increased the inflammatory response in both regions. Knockdown of cPLA2 in Sf1 neurons decreased the number of Iba1 cells in the ARC (Fig. 6l) and the size of Iba1 cells in both VMH and ARC (Fig. 6m, n). These results suggest that cPLA2 in the VMH regulates HFD-induced inflammation in the VMH and ARC. Similar results were observed in female mice (Supplementary Fig. 12).

## Discussion

The roles of hypothalamic phospholipids and eicosanoids in the regulation of energy homeostasis is ill-defined. In this study, we found that the composition of phospholipids in the hypothalamus, especially AA attached phospholipids, are dynamically affected by blood glucose levels. cPLA2 in the VMH plays an important role in AA metabolism to produce prostaglandins and

increase insulin sensitivity in skeletal muscles during hyperglycemia in RCD-fed mice. cPLA2-mediated phospholipid metabolism also regulates glucose-responsiveness in the dmVMH. HFD feeding, which promotes hyperglycemia, continuously activates cPLA2 and produces prostaglandins, and thus induces inflammation in the hypothalamus and attenuates insulin sensitivity in the liver (Fig. 7). Therefore, cPLA2-mediated phospholipid metabolism in the hypothalamus is critical for the physiological and pathological control of systemic glucose homeostasis.

FAs and PUFAs are believed to be transported from the bloodstream to the hypothalamus, and FA metabolism in the hypothalamus changes food intake and energy expenditure[19,31]. However, we observed reductions in AA-containing phospholipids, increases in lysospecies and increases in prostaglandins in the hypothalamus after glucose injection. This suggests that AA is produced from intrinsic membrane phospholipids in the hypothalamus to make eicosanoids during hyperglycemia. The produced prostaglandins play important roles in controlling hyperglycemia because the injection of COX inhibitor impaired glucose tolerance. It has been reported that FA oxidation by carnitine palmitoyltransferase I in the VMH plays important roles in food intake and energy homeostasis[32]. However, our data showed that the cPLA2 in Sf1 neurons has a minor effect on changes in body weight and tissue weight. This suggests that cPLA2 in Sf1 neurons controls glucose metabolism, but not body weight regulation, and it is likely that FAs generated from phospholipids are utilized for prostaglandin production.

AA exists in the sn-2 position of phospholipids, and cPLA2 is the rate-limiting enzyme for catalyzing AA by extracellular stimulation. cPLA2 is activated by an increase in the intracellular calcium concentration and by the phosphorylation of 505-serine residue, which is induced by the MAP kinase pathway[27]. The mechanism that activated cPLA2 in our study remains to be elucidated. However, we found that the glucose-induced activation of the dmVMH is dependent on prostaglandin production by Sf1 neurons. Sf1 neurons exist mainly in the dmVMH, and most AA-containing phospholipids were found near the third ventricle in our study. Therefore, the hyperglycemia-induced prostaglandin production occurs in the medial part of the hypothalamus and affects neuronal activity in this region probably via changes in ion channel activities[33]. Similarly, prostaglandins regulate glucose-stimulated insulin secretion (GSIS) from pancreatic beta cells[34].

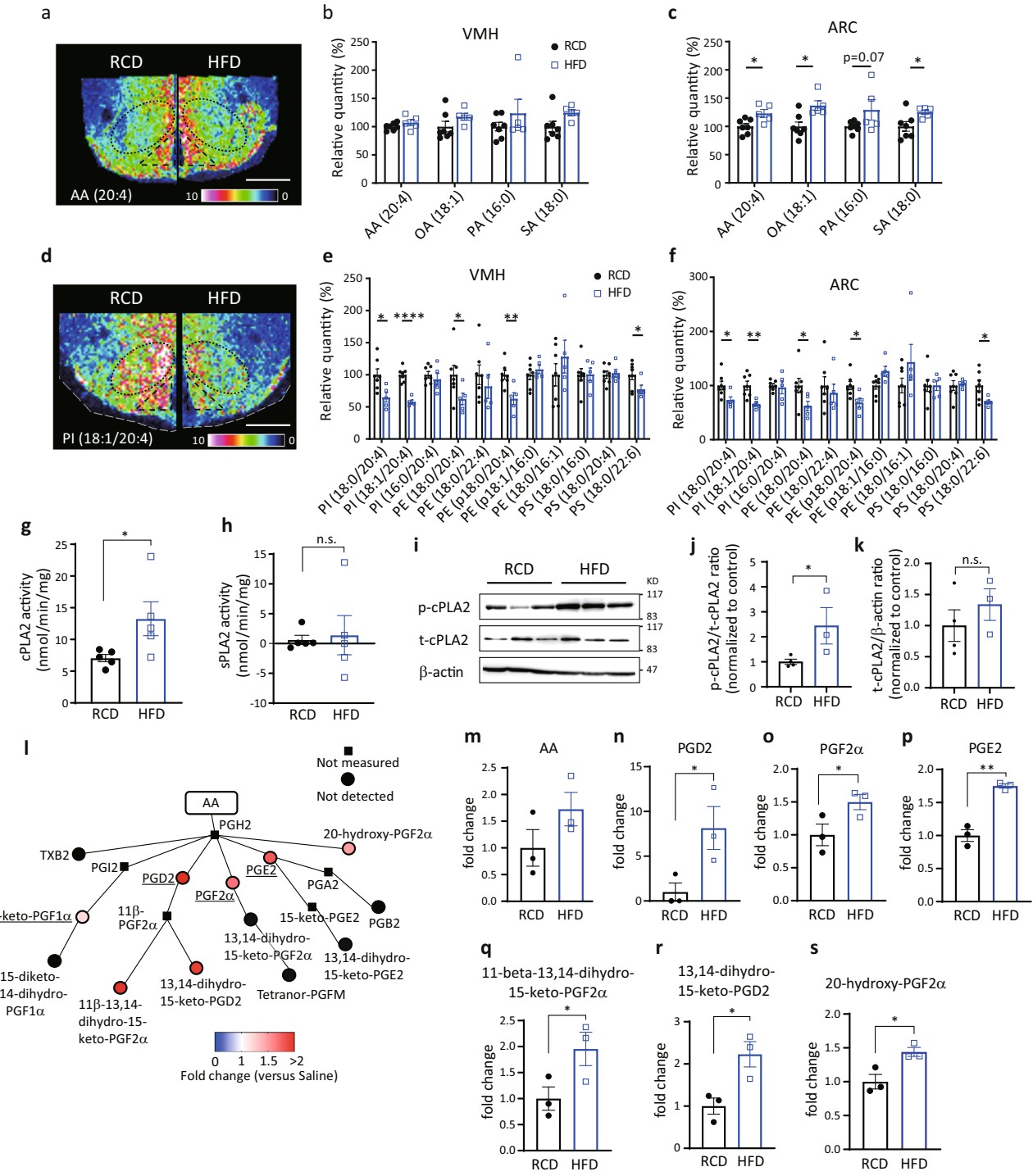

GSIS is the most studied mechanism of glucose sensing. Thus, it is possible that a similar mechanism for prostaglandins affecting GSIS may be involved in the hypothalamic glucose sensing. Neurons in the vlVMH and ARC are glucose sensing neurons[29,35]. In agreement with this, the glucose injection increased cFos-positive cells in the vlVMH and ARC. However, i.c.v. injection of inhibitors for cPLA2 and COX1/2 did not change the glucose-induced increase in cFos, suggesting that the prostaglandin production has a minor role in the glucose sensing by vlVMH and ARC. We also found that cPLA2 regulates neuronal activity of the dmVMH in response to leptin. Therefore, cPLA2 may be required

for the activation of dmVMH not only by glucose but also by peripheral hormones.

Sf1 neurons are critical for the regulation of whole-body energy homeostasis[7,9]. Activation of VMH neurons increases glucose uptake in skeletal muscle and BAT, but not in WAT or other organs[36,37]. Similar results were found in mice with intra-VMH administration of leptin[11,12,38]. Leptin receptors are located in the dmVMH and required to maintain normal glucose homeostasis[39–41]. In agreement with this, glucose sensing by Sf1 neurons via UCP2 is also critical for systemic glucose metabolism[42]. Therefore, it is plausible that the activation of the dmVMH by

**Fig. 4 HFD feeding increases prostaglandin production derived from phospholipids. a–f** Distributions of fatty acids and phospholipids in the hypothalamus in regular chew diet- (RCD-) or high-fat diet- (HFD-) fed mice for 8 weeks. **a, d** Representative results of IMS on hypothalamic arachidonic acid (AA) (**a**) and PI (18:1/20:4) (**d**) from RCD-fed mice (left) or HFD-fed mice (right). Scale bar: 500 µm. **b, c** Relative intensities of fatty acids in the VMH (**b**) or ARC (**c**) of RCD- ($n = 7$) or HFD-fed mice ($n = 5$; two-tailed $t$ test for each molecule, ARC: $p = 0.0190$ in AA(20:4), $p = 0.0104$ in OA(18:1), $p = 0.0387$ in SA(18:0), RCD vs HFD). **e, f** Relative intensities of phospholipids in the VMH (**e**) or ARC (**f**) of RCD- ($n = 7$) or HFD-fed ($n = 5$) mice (two-tailed $t$ test for each molecule, for VMH: $p = 0.0196$ in PI (18:0/20:4), $p < 0.0001$ for PI (18:1/20:4), $p = 0.0362$ for PE (18:0/20:4), $p = 0.0080$ for PE (p18:0/20:4), $p = 0.0307$ for PS (18:0/22:6), for ARC: $p = 0.0325$ for PI (18:0/20:4), $p = 0.0061$ for PI (18:1/20:4), $p = 0.0282$ for PE (18:0/20:4), $p = 0.0192$ for PE (p18:0/20:4), and $p = 0.0347$ for PS (18:0/22:6)). **g, h** Enzymatic activity of hypothalamic cPLA2 (**g**) and sPLA (**h**) in RCD- ($n = 5$) or HFD-fed ($n = 5$) mice (two-tailed $t$ test, $p = 0.0264$ in **g**, RCD vs HFD). **i–k** Relative amount of total-cPLA2 (t-cPLA2) and phosphorylated-cPLA2 (p-cPLA2) between RCD and HFD-fed mice. **i** Representative photos of western blotting. **j, k** Quantification of p-cPLA2 (**j**) and t-cPLA2 (**k**) between RCD ($n = 4$) and HFD ($n = 3$) fed mice (two-tailed $t$ test, $p = 0.0338$ in **j**, RCD vs HFD). **l** Relative amounts of prostaglandins in the hypothalamus after 8 weeks in HFD-fed mice ($n = 3$) compared with those of RCD-fed mice ($n = 3$). Major prostaglandins were underlined. **m–s** Bar graphs showing COX-mediated production of AA (**m**), PGD2 (**n**), PGF2α (**o**), PGE2 (**p**), 11-beta-13,14-dihydro-15-keto-PGF2α (**q**), 13,14-dihydro-15-keto-PGD2 (**r**), and 20-hydroxy-PGF2α (**s**) in 8 weeks of HFD-fed mice ($n = 3$) compared with RCD-fed mice ($n = 3$; two-tailed $t$ test, $p = 0.0256$ in **n**, $p = 0.0339$ in **o**, $p = 0.0013$ in **p**, $p = 0.0354$ in **q**, $p = 0.0260$ in **r**, $p = 0.0250$ in **s**, RCD vs HFD). All data represent the mean ± SEM; *$p < 0.05$; **$p < 0.01$; ***$p < 0.001$; ****$p < 0.0001$.

glucose injection regulates insulin sensitivity in skeletal muscle through cPLA2-mediated prostaglandin production.

VMH neurons are also known to evoke counter regulatory responses (CRR) during hypoglycemia[4]. We firstly thought that the fasting may affect the neuronal activity of glucose inhibited neurons in the VMH to evoke CRR. However, blood glucose levels after fasting or 2DG injection were similar between cPLA2KD$^{Sf1}$ and GFP$^{Sf1}$ mice, suggesting that the cPLA2 in the VMH has minor roles in the CRR. Thus, we started to use fasted mice in GTT because it can set the feeding condition similar in each animal.

A HFD feeding causes diet-induced-obesity and a state of chronic, low-grade inflammation occurs in several tissues, including the hypothalamus[6]. This hypothalamic inflammation is accompanied by an activation of microglia, and these changes decrease activities of POMC and AgRP neurons in response to several endocrine signals, such as leptin and insulin[2]. We also found that microglia was activated by the HFD feeding in both VMH and ARC. Additionally, a HFD feeding increases astrogliosis in the ARC, paraventrical hypothalamus and dorsomedial hypothalamus, but not the VMH[43]. Our data also shows that a HFD feeding induces astrogliosis in the ARC but not the VMH.

After the mice in this study were fed with HFD, FAs such as AA, OA, PA and SA accumulated in the hypothalamus, especially in the ARC. However, AA-containing phospholipids decreased because of an increase in hypothalamic cPLA2 activity. The increased cPLA2 activity is regulated by phosphorylation of cPLA2, but not its protein expression. Prostaglandins are proinflammatory signals in the brain[44] and knockdown of cPLA2 in Sf1 neurons attenuates inflammation in the hypothalamus. Knockdown of cPLA2 in Sf1 neurons of female mice also improved glucose metabolism and ameliorated hypothalamic inflammation, suggesting that the effects of cPLA2 in HFD-fed mice do not depend on sex in our study, even though the responses to HFD are reported to be different between male and female mice[45].

It is possible that the long-term production of prostaglandins, which has a physiological role in glucose metabolism in RCD-fed mice, initiates HFD-induced inflammation which impairs neuronal functions. A limitation of this study is that we do not know how long the prostaglandin-induced neuroinflammation last after the end of HFD feeding. This should be investigated in future because this is important for the clinical intervention to treat type II diabetes after changing the food.

It is notable that the species of PGs produced by the glucose injection in RCD-fed mice are different from PGs after HFD feeding (Figs. 1 and 4). Glucose injection increased 6-keto-PGF1α

and 13,14-dihydro-15-keto-PGF2α, while HFD feeding stimulated the production of PGF2α, 11-beta-13,14-dihydro-15-keto-PGF2α, 13,14-dihydro-15-keto-PGD2, and 20-hydroxy-PGF2α. Both glucose and HFD produced PGD2 and PGE2. It has been reported that i.c.v. injection of PGE2, PGD2, or PGF2α increases hepatic glucose production and promotes hyperglycemia in RCD-fed rodents[21,46]. Although one of the reasons is that i.c.v. injection affects the whole brain, the role of PGs is opposite to our findings, i.e., PGs decreases hyperglycemia in RCD-fed mice in our study. Therefore, hypothalamic 6-keto-PGF1α and 13,14-dihydro-15-keto-PGF2α may be important for the regulation of insulin sensitivity in the muscle in healthy mice. Many PGs promote inflammation, however, the role of each PG in the proinflammatory effects of HFD is not clear. Further studies are needed to identify the responsible PG and its receptor to regulate hypothalamic neuronal activity and inflammation. Once cPLA2 is activated, it produces AA and lysospecies, such as LPI, LPE, and LPS. AA will be used not only in the eicosanoid production but also in the β-oxidation and lipid synthesis including phospholipid remodeling, as known as Land's cycle[47,48]. The amounts of AA measured by LC-MS was increased by HFD feeding. Therefore, AA may be kept in the AA-pool in the cytosol and utilized in the membrane remodeling or β-oxidation in the hypothalamus. LPI (18:0) is known as an agonist of G-protein coupled receptor 55 (GPR55). Whole-body knock out of GPR55 develops insulin resistance[49]. However, the role of hypothalamic GPR55 is still not clear. More data is required to reveal the importance of membrane remodeling and LPI/GPR55 pathway in the hypothalamus for the systemic glucose metabolism.

We also found that a HFD feeding abolished the increase in cFos expression induced by glucose in the VMH and ARC, which is consistent with the report that HFD feeding decreases glucose sensing by VMH and POMC neurons[28,29]. In the present study, knockdown of cPLA2 improved the glucose response in the vlVMH and ARC in HFD-fed mice. However, knockdown of cPLA2 had no effect on the glucose responsiveness of the dmVMH in HFD-fed mice. We also observed that knockdown of cPLA2 impaired glucose sensing in the dmVMH in RCD-fed mice. This indicates that the same attenuation of glucose sensing has already occurred in the dmVMH of HFD-fed mice, thus the dmVMH could not respond to the glucose injection in HFD-fed cPLA2KD$^{Sf1}$ mice. Although Sf1 neurons exist mainly in the dmVMH[39], the prostaglandin production originated from the dmVMH may enhance inflammatory responses in the whole hypothalamus, including the ARC and vlVMH. The vlVMH is filled with glucose-sensing neurons, both glucose-excited and glucose-inhibited neurons, and regulated systemic glucose balance[35]. Our results show that HFD feeding inhibited neuronal glucose-sensing in the vlVMH. This impairment

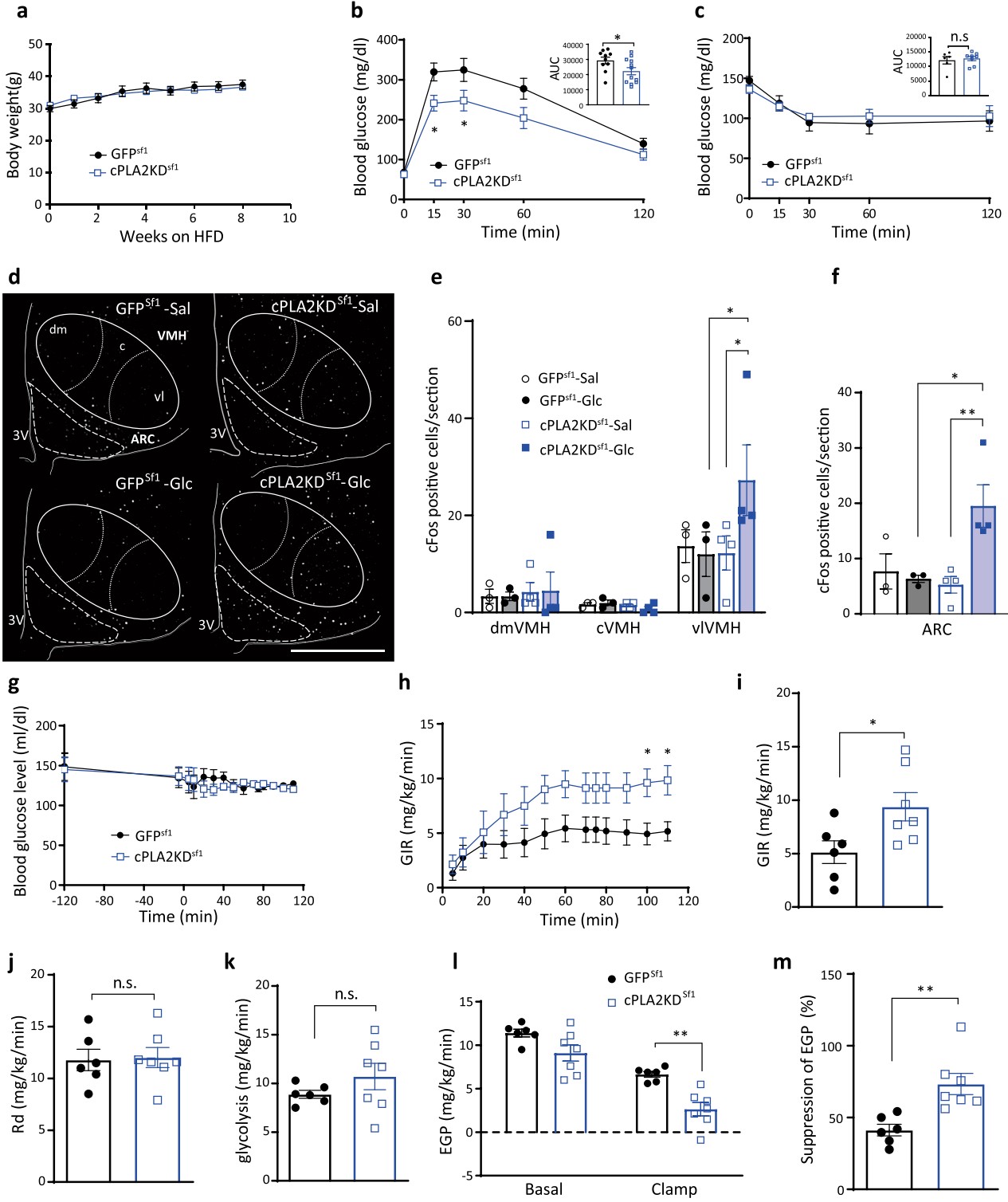

of glucose-sensing in vlVMH was recovered in cPLA2KD[sf1] mice accompanied with ameliorated hypothalamic inflammation. Therefore, the cPLA2 may play a deteriorative role in the glucose responsiveness of the VMH/ARC by inducing hypothalamic inflammation. In the present study, we did not knockdown cPLA2 in neurons of the ARC and accumulations of fatty acids were observed in the ARC after HFD feeding. Thus, the importance of PG production and lipid synthesis in the ARC on inflammation is not clearly defined. Further study is needed to understand the

whole mechanism which initiates and deteriorates hypothalamic inflammation.

Our data suggest that inflammation of the hypothalamus contributes to attenuating glucose sensing by the VMH and ARC. POMC and AgRP neurons are reported to regulate hepatic insulin sensitivity, but not muscle glucose metabolism[14,15,50]. Therefore, the improvement of the neuronal activity in the ARC contributes to restoring glucose metabolism by changing insulin sensitivity in the liver.

**Fig. 5 Knockdown of cPLA2 improves HFD-induced impairment of glucose metabolism. a** Body weight change in cPLA2KD[Sf1] mice ($n = 12$) and GFP[Sf1] mice ($n = 11$). **b** Glucose tolerance test on cPLA2KD[Sf1] mice ($n = 12$) and GFP[Sf1] mice ($n = 10$; two-way ANOVA followed by Sidak multiple comparison test, $p = 0.0415$ at time = 15, $p = 0.0482$ at time = 30, cPLA2KD[Sf1] vs GFP[sf1]; two-tailed t test in area under the curve (AUC) during GTT, $p = 0.0428$, cPLA2KD[Sf1] vs GFP[sf1]). **c** Insulin tolerance test on cPLA2KD[Sf1] ($n = 8$) mice and GFP[Sf1] mice ($n = 6$) during 8 weeks of HFD feeding. **d** Representative micrographs showing immunofluorescent cFos staining in the hypothalamus of HFD-fed cPLA2KD[Sf1] and GFP[Sf1] mice after saline or glucose injection. Scale bar: 500 μm. **e, f** Quantification of cFos expression in the dmVMH, cVMH, vlVMH (**e**) and ARC (**f**) of HFD-fed cPLA2KD[Sf1] or GFP[Sf1] mice after saline or glucose injection ($n = 3$-4 in each experimental group; VMH: two-way ANOVA followed by Sidak multiple comparison test. for vlVMH: $p = 0.0231$ cPLA2KD[Sf1] Sal vs cPLA2KD[Sf1] Glc, $p = 0.014$ GFP[Sf1] Glc vs cPLA2KD[Sf1] Glc. For ARC: one-way ANOVA followed by Sidak multiple comparison test, $p = 0.0176$ cPLA2KD[Sf1] Sal vs cPLA2KD[Sf1] Glc, $p = 0.0444$ GFP[Sf1] Glc vs cPLA2KD[Sf1] Glc). **g**–**m** Hyperinsulinemic–euglycemic clamp studies in HFD-fed cPLA2KD[Sf1] ($n = 7$) or GFP[Sf1] ($n = 7$) mice. **g** Blood glucose levels during hyperinsulinemic–euglycemic clamp studies in HFD-fed cPLA2KD[Sf1] ($n = 7$) or GFP[Sf1] ($n = 7$) mice. **h** The glucose infusion rate (GIR) required to maintain euglycemia during the clamp period in cPLA2KD[Sf1] ($n = 7$) or GFP[Sf1] ($n = 6$) mice. **i** The average GIR between 75 and 115 min in cPLA2KD[Sf1] ($n = 7$) or GFP[Sf1] ($n = 6$) mice (two-tailed t test, $p = 0.0324$, cPLA2KD[Sf1] vs GFP[Sf1]). **j** The rate of glucose disappearance (Rd) during the clamp period, which represents whole-body glucose utilization. **k** The rates of whole-body glycolysis in cPLA2KD[Sf1] or GFP[Sf1] mice. **l** Endogenous glucose production (EGP) during both basal and clamp periods in cPLA2KD[Sf1] or GFP[Sf1] mice. **m** The percent-suppression levels of EGP induced by insulin infusion, which represents hepatic insulin sensitivity in cPLA2KD[Sf1] ($n = 7$) or GFP[Sf1] ($n = 6$) mice (two-tailed t test, $p = 0.0038$, cPLA2KD[Sf1] vs GFP[Sf1]). All data represent the mean ± SEM; $*p < 0.05$; $**p < 0.01$.

Aspirin, a COX inhibitor, suppresses insulin sensitivity in healthy human[51–53], but improves insulin resistance in diabetic patients[54]. Our results were in a good agreement with the human studies. The hypothalamic prostaglandin production may be critical for the effects of aspirin on the whole-body insulin sensitivity.

In summary, our study shows that the cPLA2 is fundamental for the function of the hypothalamus in regulating glucose homeostasis. Neuronal cPLA2 is necessary for their own activities in the dmVMH to respond to glucose and control blood glucose levels. However, cPLA2 in the VMH also has the critical role in inducing hypothalamic inflammation during HFD feeding. Therefore, the role of cPLA2-mediated eicosanoid production in the hypothalamus is different between RCD and HFD. Our findings provide evidence that cPLA2-mediated phospholipid metabolism in hypothalamic neurons plays an important role in systemic glucose metabolism.

## Methods

**Reagents**. All the reagents and resources used in this study are listed in the Supplementary table 1.

**Animals**. Sf1-cre mice were purchased from the Jackson Laboratory (STOCK Tg (Nr5a1-cre)7Lowl/J; Bar Harbor, ME). For IMS and assessing the effects of inhibitors, male C57BL6J mice were purchased from Charles River Laboratories Japan. All mice were kept at 22–24 °C and 30–60% humidity with a 12-h light/12-h dark cycle and given ad libitum food access. Mice were fed RCD (Nosan Corporation, Yokohama, Japan) or 45% HFD (D12451; Research Diet Inc., New Brunswick, NJ). Animal care and experimental procedures were performed in accordance with guidelines and approval from the Animal Care and Use Committee of Hokkaido University.

**Imaging mass spectrometry**. Glucose (2 g/kg body weight, Sigma-Aldrich, St. Louis, MO) or saline were injected intraperitoneally (i.p.) and the mice were killed using CO2 asphyxiation 30 min after injection. Brains were immediately placed into ice-cold saline and were embedded in 2% sodium carboxymethyl cellulose solution and frozen with liquid nitrogen. The 10-μm brain sections were prepared by cryostat and immediately mounted onto an indium-tin-oxide-coated glass slide (Bruker Daltonics, Bremen, Germany). The sections on the glass slides were immediately dried and stored at −20 °C until imaging mass spectrometry analysis.

Brain slices were sprayed with 9-aminoacridine matrix (10 mg/mL in 70% ethanol, Sigma-Aldrich) and installed into a matrix-assisted laser desorption/ionization (MALDI)-time-of-flight (TOF)/TOF system using ultrafleXtreme (Bruker Daltonics). Brain sections were irradiated by a smart beam (Nd:YAG laser, 355-nm wavelength); with a 25-μm irradiation pitch. The laser had a repetition frequency of 2000 Hz and mass spectra were obtained in the range of $m/z$ 200–1200 in negative-ion mode. The $m/z$ values from previous reports were used to label each lipid, phospholipid and lysospecies (Supplementary table 2)[55–58]. All ion images were reconstructed with total ion current (TIC) normalization by flexImaging (Bruker Daltonics) and transferred to ImageJ after modifying the grayscale. The areas of the VMH were identified by DAPI staining in the other brain sections, and the brightness of mass spectrometry signals in the VMH and ARC were calculated

as intensity. Phosphatidyl-choline and phosphatidyl-glycerol were measured in the adjacent tissue slices with positive-ion mode. However, the mass spectra after MS/MS were not sufficiently conclusive to allow a reliable data assignment, especially the fatty acid components.

**Quantification of PGs**. Glucose (2 g/kg body weight, Sigma-Aldrich) or saline was injected i.p. two times ($t = 0$ and $t = 30$ min) and mice were killed using CO2 asphyxiation at $t = 60$ min. A HFD (45 kcal% fat, D12451, Research Diet, NJ) was given for 8 weeks. The hypothalamus was collected and immediately frozen in liquid nitrogen. The tissue was homogenized with 500 μl of MeOH:formic acid (100:0.2) containing an internal standard consisting of a mixture of deuterium-labeled PGs using microtip sonication. The samples were submitted to solid phase extraction using an Oasis HLB cartridge (5 mg; Waters, Milford, MA) according to the method of Kita et al[59]. Briefly, samples were diluted with water:formic acid (100:0.03) to give a final MeOH concentration of ~20% by volume, applied to preconditioned cartridges, and washed serially with water:formic acid (100:0.03), water:ethanol:formic acid (90:10:0.03), and petroleum ether. Samples were eluted with 200 μl of MeOH:formic acid (100:0.2). The filtrate was concentrated with a vacuum concentrator (SpeedVac, Thermo Fisher Scientific, Waltham, MA). The concentrated filtrate was dissolved in 50 μL of methanol and used for liquid chromatography/mass spectrometry (LC-MS).

The PGs in the hypothalamus were quantified by liquid chromatography/mass spectrometry based on the method described in Yamada et. al.[60]. Briefly, a triple-quadrupole mass spectrometer equipped with an electrospray ionization (ESI) ion source (LCMS-8040, Shimadzu Corporation) was used in the positive and negative-ESI and selective reaction monitoring modes. The samples were resolved on a reversed-phase column (Kinetex C8, 2.1 × 150 mm, 2.6 μm, Phenomenex, Torrance, CA) using a step gradient with mobile phase A (0.1% formate) and mobile phase B (acetonitrile). The gradient of mobile phase B concentration was programmed as 10% (0 min)−25% (5 min)−35% (10 min)−75% (20 min)−95% (20.1 min)−95% (28 min)−10% (28.1 min)−10% (30 min), at a flow rate of 0.4 mL/min and a column temperature of 40 °C. Conditions of Multiple Reaction Monitoring (MRM) for each PGs are shown in the supplementary table 3.

**Stereotaxic surgeries and AAV injection**. Male C57BL/6J mice were anesthetized with mixture of ketamine (100 mg/kg) and xylazine (10 mg/kg) and were put on a stereotaxic instrument (Narishige, Tokyo, Japan). Mice were implanted with cannulae for intracerebroventricular (i.c.v.) or intra-hypothalamic injection. The i.c.v. cannulae were implanted in the lateral ventricle in an anterior–posterior (AP) direction: −0.3 (0.3 mm posterior to the bregma), lateral (L): 1.0 (1.0 mm lateral to the bregma), dorsal–ventral (DV): −2.5 (2.5 mm below the bregma on the surface of the skull). The double-cannulae for intra-hypothalamic injection had a gap of 0.8 mm between the two cannulae and were implanted following the coordinates of the AP: − 1.4, L: ± 0.4, DV: − 5.6. Cannulae were secured on the skulls with cyanoacrylic glue and the exposed skulls were covered with dental cement. To knockdown expression of *pla2g4a* in the VMH, 6- to 8-week-old Sf1-cre mice were injected in each side of the VMH with ~0.5 μL AAV8-DIO-shRNA (Vigene Biosciences, Rockville, MD) against *mpla2g4* bilaterally using the following coordinates: AP: − 1.4, L: ± 0.4, DV: − 5.7. Open wounds were sutured after viral injection. Mice were allowed to recover for 5–7 days before experiments were started. Brains were collected to check the injection site (see the section of "Immunohistochemistry" below). The mice, in which the AAV injection was not successful, were removed from the data.

**Measurement of blood glucose levels after glucose, insulin, 2DG or refeeding**. A glucose tolerance test was performed on ad libitum fed or fasted mice. The ad

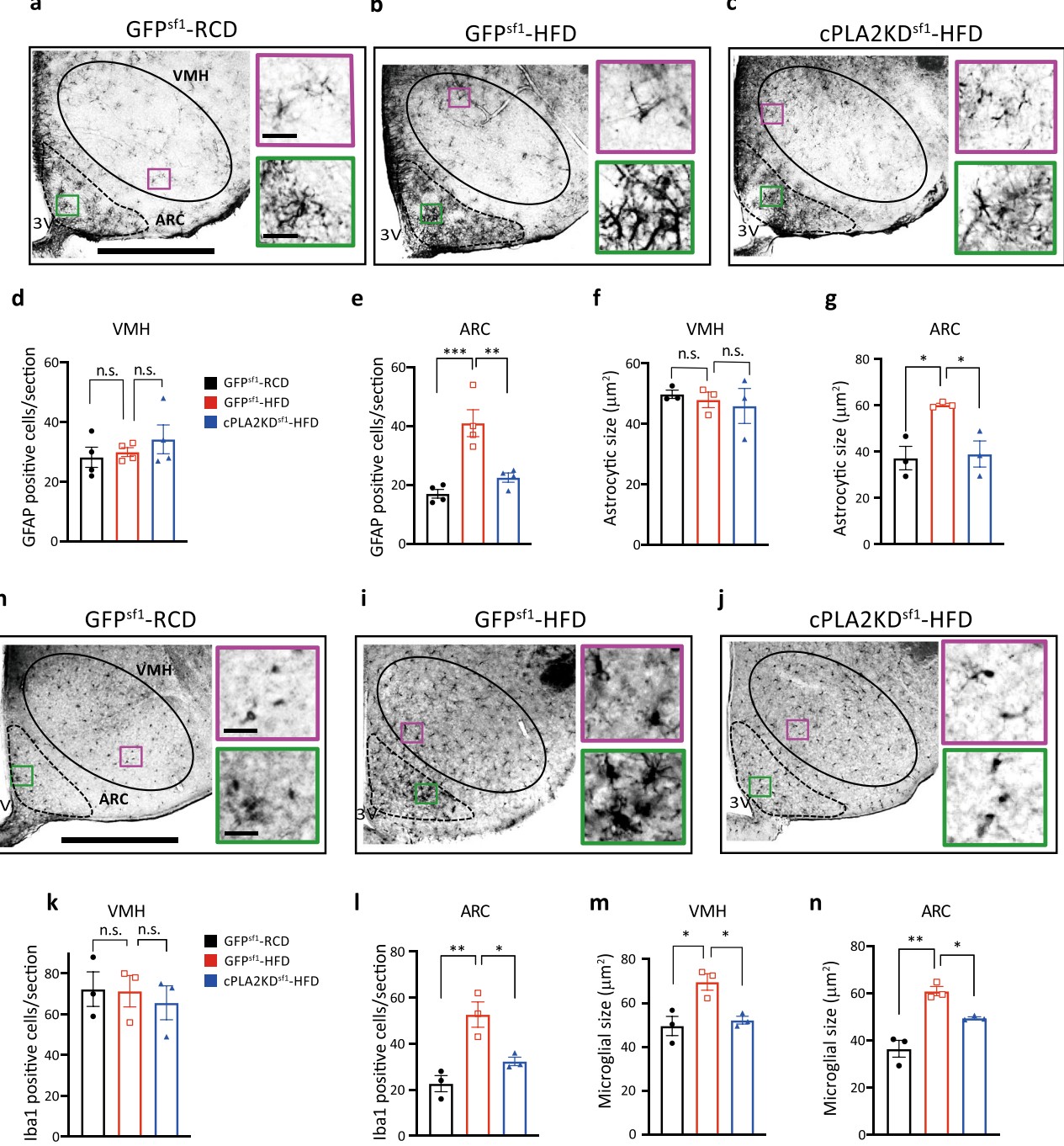

**Fig. 6 Knockdown of cPLA2 prevents HFD-induced microgliosis and astrogliosis. a–c** Left: representative micrographs showing immunochemistry GFAP staining in the hypothalamus of RCD-fed GFP[Sf1] mice (GFP[Sf1]-RCD) (**a**), HFD-fed GFP[Sf1] mice (GFP[Sf1]-HFD) (**b**), and HFD-fed cPLA2KD[Sf1] (cPLA2KD[Sf1]-HFD) mice (**c**). Scale bar: 500 μm. Right: magnified areas in the VMH and ARC in the left. Scale bar: 30 μm. d,e, Quantification of GFAP-positive cells in the VMH (**d**) or ARC (**e**) of GFP[Sf1]-RCD ($n = 4$), GFP[Sf1]-HFD ($n = 4$) and cPLA2KD[Sf1]-HFD ($n = 4$) mice (one-way ANOVA followed by Sidak multiple comparison test, in **e**, $p = 0.0007$ GFP[Sf1] RCD vs GFP[Sf1] HFD, $p = 0.0045$ GFP[Sf1] HFD vs cPLA2KD[Sf1] HFD). **f,g,** Size of GFAP-positive cells in in the VMH (**f**) or ARC (**g**) of GFP[Sf1]-RCD ($n = 3$), GFP[Sf1]-HFD ($n = 3$), and cPLA2KD[Sf1]-HFD ($n = 3$) mice (one-way ANOVA followed by Sidak multiple comparison test, in **g**, $p = 0.0002$ GFP[Sf1] RCD vs GFP[Sf1] HFD, $p = 0.0012$ GFP[Sf1] HFD vs cPLA2KD[Sf1] HFD). **h–j** Left: Representative micrographs showing immunochemistry Iba1 staining in the hypothalamus of GFP[Sf1]-RCD (**h**), GFP[Sf1]-HFD (**i**) and cPLA2KD[Sf1]-HFD mice (**j**). Scale bar: 500 μm. Right: magnified areas in the VMH and ARC from the left photos. Scale bar: 30 μm. **k, l** Quantification of Iba1-positive cells in the VMH (**k**) or ARC (**l**) of GFP[Sf1]-RCD ($n = 3$), GFP[Sf1]-HFD ($n = 3$), and cPLA2KD[Sf1]-HFD ($n = 3$) mice (one-way ANOVA followed by Sidak multiple comparison test, in **l**, $p = 0.0049$ GFP[Sf1] RCD vs GFP[Sf1] HFD, $p = 0.0310$ GFP[Sf1] HFD vs cPLA2KD[Sf1] HFD). **m, n** Size of Iba1-positive cells in the VMH (**m**) or ARC (**n**) of GFP[Sf1]-RCD ($n = 3$), GFP[Sf1]-HFD ($n = 3$), and cPLA2KD[Sf1]-HFD ($n = 3$) mice (one-way ANOVA followed by Sidak multiple comparison test, in **m**, $p < 0.0001$ GFP[Sf1] RCD vs GFP[Sf1] HFD, $p = 0.0004$ GFP[Sf1] HFD vs cPLA2KD[Sf1] HFD, in **n**, $p < 00001$ GFP[Sf1] RCD vs GFP[Sf1] HFD, $p = 0.0004$ GFP[Sf1] HFD vs cPLA2KD[Sf1]). All data represent the mean ± SEM; *$p < 0.05$; **$p < 0.01$; ***$p < 0.001$.

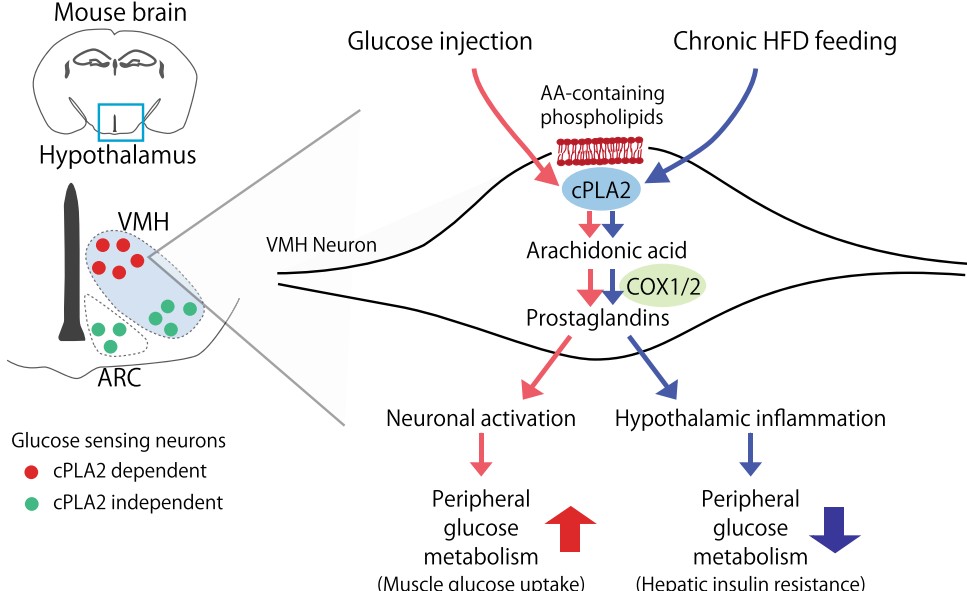

**Fig. 7 Distinct roles of prostaglandin in the regulation of peripheral glucose metabolism.** In RCD-fed mice, a glucose injection activates cPLA2 in the VMH, which increases the production of prostaglandins from neurons. The increase in prostaglandins is critical for the activation of dmVMH and glucose metabolism in peripheral tissues. Chronic HFD feeding also increases the cPLA2-mediated prostaglandin production from VMH neurons. The chronic effect of prostaglandin enhances hypothalamic inflammation and thus impairs peripheral glucose metabolism.

libitum fed mice were used for assessing the effects of inhibitors. The fasted mice were used for assessing the phenotype of mice with knockdown of *pla2g4a* in Sf1 neurons (cPLA2KD$^{Sf1}$). To assess the effects of inhibitors, methyl arachidonyl fluorophosphonate (MAFP; 20 μM, 300 nL in each side), indomethacin (140 μM, 300 nL in each side), or vehicle were injected into both sides of the hypothalamus through a double-cannula. Glucose solution was then injected i.p. (2 g/kg) 30 min after intra-hypothalamic injection. To assess the phenotype of cPLA2KD$^{Sf1}$ mice, animals were fasted for 16 h and injected with glucose (2 g/kg) i.p. During GTT, ITT and 2DG-induced glucose deprivation, a small drop of blood (2 μl) from the tail vein after poking the tail tip with a needle was placed to a strip of the handheld glucometer (Nipro Free style, Nipro, Osaka, Japan) to measure the blood glucose level. Blood glucose levels were measured before injecting inhibitors (−30 min) or glucose (0 min), and measured at 15, 30, 60, and 120 min after glucose injection.

An insulin tolerance test and the 2DG-induced glucose deprivation was performed in ad libitum fed mice. The mice were i.p. injected with 0.5 U/kg insulin (Novo Nordisk, Bagsværd, Denmark) or 400 mg/kg 2DG (Sigma-Aldrich). Blood glucose was measured at 0, 15, 30, 60, and 120 min after the injection.

For assessing blood glucose levels after refeeding, animals were fasted for 16 h and given RCD (20 g food per kg body weight). Blood glucose was measured at 0, 30, 60, 90, and 120 min.

**Serum insulin measurement.** Mice were injected with inhibitors or vehicle into the hypothalamus using the same protocol as above. Blood (40 μl) from the tails was taken 30 min after intra-hypothalamic injection. Then, glucose (2 g/kg) was i.p. injected and blood samples (40 μl) from the tail were taken at 15 and 30 min after glucose injection. The serums were collected after centrifuging for 10 min at 1000×g and maintained at −80 °C until insulin was measured. The insulin concentration was measured with a Mouse Insulin ELISA KIT (FUJIFILM Wako, Osaka, Japan) and all procedures were followed by the protocols provided in the kit.

**Real-time PCR.** Total RNA was extracted from the whole hypothalamus using Trizol solution (Invitrogen). A high capacity cDNA reverse transcription kit (Thermo Fisher Scientific) was used for the reverse transcription. *Pla2g4*, *Rbfox3*, and *Actb* cDNA levels in the hypothalamus were measured by TaqMan Gene Expression Assays (Thermo Fisher Scientific, assay ID: *Pla2g4*, Mm00447040_m1; *Rbfox3*, Mm01248771_m1; *Actb*, Mm02619580_g1). Real-time PCR (LightCycler 480; Roche) was performed with diluted cDNAs in a 20-μl reaction volume in triplicate.

**Immunohistochemistry.** Ad libitum fed mice were i.p. injected with either saline or glucose (3 g/kg), killed using CO2 asphyxiation and perfused with heparinized saline followed by 4% paraformaldehyde (PFA) transcardially at 30 min after injection. Inhibitors were i.c.v. injected 30 min before glucose injection. To assess the effect of MAFP on leptin signals, leptin was injected i.c.v. 10 min after i.c.v. injection of MAFP. Mice were killed 30 min after leptin injection and perfused with

saline and 4% PFA as mentioned above. Brain sections (50 μm each) containing the whole VMH were collected. For cFos and GFP staining, the floating sections were incubated with rabbit-anti-cFos antibody (1:200, Santa Cruz Biotechnology, Denton, TX) or rabbit-anti-GFP antibody (1:1000, Frontier Institute, Hokkaido, Japan) in staining solution (0.1 M phosphate buffer (PB) containing 4% normal guinea pig serum, 0.1% glycine, and 0.2% Triton X-100) overnight at room temperature. After rinsing with PB, sections were incubated in secondary antibody (1:500, Alexa 488 or 647 Goat Anti-Rabbit (IgG) secondary antibody, Cell Signaling Technologies, Danvers, MA) for 2 h at room temperature. For staining of phosphorylated STAT3 (pSTAT3), antigen was retrieved by incubated with 10 mM citrate sodium at 80 °C for 30 min. Sections were then incubated with Rabbit-anti-pSTAT3 (Tyr 705) antibody (Cell Signaling Technologies) in staining solution overnight at 4 °C. After rinsing with PB, sections were incubated with biotinylated-Goat-Anti-Rabbit (IgG) secondary antibody (1:500, Thermo Fisher Scientific) for 1 h followed by incubated in Alexa 488-streptavidin for 2 h at room temperature. The stained sections were washed with PB three times and mounted on glass slides with vectashield (Vector Laboratories, Burlingame, CA).

To assess the cell population of astrocytes and microglia, sections were treated with 1% H$_2$O$_2$ for 20 mins followed by 20 min treatment of 0.1% Triton-X 100 at room temperature. The sections were washed with PB three times and incubated with rabbit-anti-Iba1 antibody (1:3000, FUJIFILM Wako) or rabbit-anti-GFAP antibody (1:3000, Sigma-Aldrich) in staining solution overnight at room temperature. After rinsing with PB, sections were incubated in biotinylated Goat Anti-Rabbit (IgG) secondary antibody (1:500, Thermo Fisher Scientific) for 1 h. Sections were then treated with ABC solution (Vector Laboratories) and incubated in DAB substrate (FUJIFILM Wako) for 4 mins after rinse with PB. To assess cell numbers (cfos-, pSTAT3-, Iba1, and GFAP-positive cells), cells were manually counted from 3 hypothalamic level-matched section containing VMH and ARC per animal using ImageJ software. Cell size (Iba1- and GFAP-positive cells) in VMH and ARC, a total of 10–13 cells in each nucleus per section, was measured using ImageJ.

**Assessment of cytosolic- or secretory-phospholipase-A2 activity.** Mice fed ad libitum were i.p. injected with either glucose (2 g/kg) or saline. Mice were killed using CO2 asphyxiation and mouse hypothalami were collected 30 min after injection and stored at −80 °C until use. Tissues were homogenized and centrifuged at 10,000×g for 15 min at 4 °C and supernatants were collected. Activity of cytosolic- or secretory-phospholipase-A2 were measured following procedures described in the kit manuals (Abcam, Cambridge, UK).

**Immunoblotting.** Eight-weeks-old male B57CL6/J mice were started to fed RCD or HFD for 2 months and killed by CO2 asphyxiation. In all, 1 mm of brain coronal sections containing VMH and ARC were made with stainless blades. The medio-basal hypothalamus (VMH and ARC) was then cut and frozen in liquid nitrogen immediately until experiments. Tissues were homogenized in PBS containing 1% Nonidet P-40 at 4 °C. The supernatant after centrifugation was fractionated by

SDS-PAGE and proteins were transferred to a polyvinylidene fluoride membrane (Immobilon; Millipore, Tokyo, Japan). The membrane was then incubated with 1ug/mL of rabbit-anti total-cPLA2 (Cell Signaling Technology) or phosphorylated (Ser505)-cPLA2 (Cell Signaling Technology) antibody and mouse-anti-β-actin antibody (Cell Signaling Technology) at 4 °C overnight. Immune complexes were visualized with horseradish peroxidase-linked goat anti-rabbit immunoglobulin and enhanced chemiluminescence reagents (GE Healthcare, Tokyo, Japan).

**Implantation of artery and vein catheter for clamp studies**. Mice were anesthetized with pre-mixed ketamine (100 mg/kg) and xylazine (10 mg/kg). Polyethylene catheters were implanted into right carotid arteries and jugular veins. The tubes entered subcutaneously and protruded from the neck skin. Mice were allowed to recover for 3–5 days and tubes were flushed with heparinized saline each day.

**Hyperinsulinemic–euglycemic clamp and measurement of 2-[$^{14}$C] deoxy-D-glucose (2DG) uptake**. The hyperinsulinemic–euglycemic clamp protocol was followed as described in previous papers[41,61]. The mice were fasted for 4 h and experiments were initiated in a free moving condition.

A 115-min clamp period ($t = 0–115$ min) was following a 90-min basal period ($t = -90$ to 0 min). A bolus of [3-$^3$H] glucose (5 mCi;) was injected through the jugular vein at the beginning of the basal period ($t = -90$ min) and tracer was infused at a rate of 0.05 mCi for 90 min. Blood samples were collected at $t = -15$ and $-5$ min to measure the rate of appearance (Ra). The clamp period was initiated with continuous infusion of insulin (2.5 mU/kg/min). During the clamp period, blood was collected and blood glucose levels were measured from arterial blood every 5–10 min. Cold glucose was infused at a variable rate via the jugular vein catheter to maintain a blood glucose level at 110–130 mg/dL. Erythrocytes in withdrawn blood were suspended in sterile saline and returned to each animal.

To assess 2DG uptake, 2-[$^{14}$C] DG (10 mCi) was infused at $t = 70$ min and blood samples were collected at $t = 75, 85, 95, 105,$ and 115 min. After collecting the blood sample at $t = 115$ min, mice were euthanatized by an intravenous infusion of thiopental sodium (Nipro, Osaka, Japan), and small pieces of tissue samples from the soleus, Gastro-R (red portion of gastrocnemius), Gastro-W (white portion of gastrocnemius), BAT, heart, spleen, EWAT (epididymal white adipose tissue), brain (cortex), and liver were rapidly collected. The rate of disappearance (Rd), which reflects whole-body glucose utilization, rate of appearance (Ra), which mainly reflects endogenous glucose production (EGP), and the rates of whole-body glycolysis and glycogen synthesis were determined as described previously[61]. In the steady state, the Ra is equal to the Rd. At first, Rd was calculated by the ratio of $^3$H-glucose infusion rate (dpm/min) to the specific activity of plasma glucose (dpm/mg), which is a ratio of plasma $^3$H-glucose (dpm/ml) to glucose levels (mg/dl). In the clamp period, Ra is composed of EGP and GIR. Thus, EGP is calculated by subtracting GIR from Rd. Whole-body glycolysis was calculated by using plasma $^3$H$_2$O. EGP is a value showing glucose production (gluconeogenesis and glycogenolysis) from the liver to the circulation, but it also includes gluconeogenesis from the kidney and other tissues. Insulin can suppress hepatic glucose production. Therefore, the suppression of EGP by insulin (Figs. 3k and 5m) mainly represents the hepatic insulin sensitivity.

**Statistical analysis and reproducibility**. Two-way or one-way ANOVA were used to determine the effect of inhibitors or knockdown of cPLA2 with the Prism 8 software (GraphPad). For repeated-measures analysis, ANOVA was used when values over different times were analyzed, followed by the Sidak multiple comparisons tests. When only two groups were analyzed, statistical significance was determined by the unpaired Student's $t$ test (two-tailed $p$ value). A value of $p < 0.05$ was considered statistically significant. All data are shown as mean ± SEM. Representative image was from at least three independent experiments.

**Reporting summary**. Further information on research design is available in the Nature Research Reporting Summary linked to this article.

## Data availability
Source data are provided with this paper. Any data not included in the source data file is available upon request.

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

## Acknowledgements

This work was supported by Leading Initiative for Excellent Young Researchers (from MEXT); a Grant-in-Aid for Young Scientists (A) (Grant Number JP17H05059), a Grant-in-Aid for Scientific Research (B) (Grant Number JP18H02857); Japanese Initiative for Progress of Research on Infectious Diseases for Global Epidemics (JP17fm0208011h0001, JP18fm0208011h0002, JP19fm0208011h0003); the Takeda Science Foundation; the Uehara Memorial Foundation; Astellas Foundation for Research on Metabolic Disorders; Suzuken Memorial Foundation; Program for supporting introduction of the new sharing system (JPMXS0420100617, JPMXS0420100618, JPMXS0420100619) and National Institutes of Health (RO1 DK107293). Infrastructure of LC-MS was supported by JST ERATO Suematsu Gas Biology Project. M.S. is the lead until March 2015.

## Author contributions

C.T. conceived this study and designed the experiments. M.L. performed most of the experiments and C.T. supervised the entire study. H.M. performed the study on astrocytes. Y.S. performed LC-MS measurements. T.H. and T.I. performed imaging mass spectrometry. I.Y. and D.I. performed GTT, ITT, and AAV injections. M.S. N.I., K.K., and S.D. assisted in preparing the manuscript.

## Competing interests

The authors declare no competing interests.
