## [Peer Review File · Nature Communications]

Reviewer comments first round –

Reviewer #1 (Remarks to the Author):

This study investigated the roles of prostaglandin (PG) production in the ventral hypothalamus in glucose homeostasis by using a variety of techniques including imaging mass spectrometry, c-Fos immunohistochemistry, intra-hypothalamic drug injections to inhibit local production of PGs, and genetic knockdown of cPLA2 mRNA selectively in Sf1-expressing neurons. The findings indicate that products of cPLA2, presumably prostaglandins, function for glucose homeostasis under normal, healthy conditions, while these products exert deteriorating effects on glucose homeostasis under high fat diet (HFD) feeding. The findings are novel and important in understanding the hypothalamic mechanisms of the regulation of glucose metabolism, and would also be relevant to diabetes medication. The experiments presented are overall fair. The authors should take care of many, but relatively minor concerns and corrections:

1. In this study, it is very important to make sure that the AAV injected into the VMH did not diffuse into the ARC and that no cPLA2 was knocked down in the ARC, because the VMH and ARC are close each other. It should be explained in the methods how this was confirmed in every mouse injected with the AAVs. Presenting an overlaid drawing that delineates infected hypothalamic areas in all the mice would be helpful to make sure this point.
2. HFD feeding increased cPLA2 activity per mg hypothalamic protein (Fig. 4g). As suggested in the discussion, HFD feeding might have promoted phosphorylation of cPLA2 to increase the catalytic activity, but it is also possible that the expression of cPLA2 was increased. Determining which is the case seems to be important in understanding how HFD stimulates tissue inflammation in the hypothalamus.
3. Hypothalamic contents of several PGs are changed after glucose infusion (Fig. 1i) and after HFD feeding (Fig. 4i). Some PGs were commonly increased, while some others were not. The PGs responsible for glucose homeostasis in healthy mice and for the proinflammatory effect of HFD may be different. It would be interesting to discuss the authors' speculation on the responsible PGs (and responsible PG receptor subtypes, if possible).
4. The glucose infusion in this study dosed the mice with 2 g/kg body weight. What physiological condition was mimicked by this dose? Is this considered physiological mimicking diet-induced hyperglycemia? Or more pathological?
5. The methods section lacks the information on the procedure of blood sampling and of anesthesia for tissue sampling and decapitation. These are essential information.
6. Line 51: Should read "Obesity can attenuate the function of these nuclei and promote...", if this statement is not applicable to all obesity conditions.
7. Line 87: Should read "Prostaglandins produced from phospholipids..."
8. Line 170: Should read: "...were co-injected into the hypothalamus of wild type mice to knock down..."
9. Line 178: How to calculate the rate of disappearance should be explained briefly. It is only referred to ref # 53 in the methods, but it is insufficient for readers to understand the result.
10. Line 220: Should read "...we fed cPLA2KD/Sf1 mice with HFD for 8 wk..."
11. Line 223: Should read "Unlike RCD-fed mice (Fig. 3b), knockdown of ..."
12. Lines 225-226: If this study was not the first to show that HFD-feeding eliminates the responses of VMH or ARC neurons to glucose, please cite the earlier paper.

13. Lines 228-229: Should read "...suggesting that cPLA2 knockdown improved neuronal responsiveness to glucose under the HFD-fed condition (Fig. 5d-f)."
14. Lines 231-232: Should read "To maintain euglycemia, significantly higher GIR was required in cPLA2KD/Sf1-HFD..."
15. Lines 231-233: For readers outside the field, it should be briefly explained why GIR was lower in HFD-fed mice (Fig. 5h, black) than RCD-fed mice (Fig. 3d, black).
16. Line 233: Should read "Unlike RCD-fed mice (Fig. 3f,g), glucose..."
17. Lines 237-238: The wording "...attenuates hepatic insulin sensitivity during HFD-induced obesity" sounds problematic for two reasons: 1) the conducted experiments did not examine insulin sensitivity exclusively in the liver, but insulin sensitivity in skeletal muscles, for example, might be altered as well; 2) the HFD-fed mice in this study did not exhibit obesity.
18. Line 261: Should read "in skeletal muscles".
19. Line 265: Is there any evidence that HFD feeding attenuated insulin sensitivity exclusively in the liver?
20. Line 297: Should read "...and are required to..."
21. Lines 310-311: What FAs are meant by "all the FAs"?
22. Line 339: Should read "...has a critical role in inducing..."
23. Line 374: brightness of WHAT? Brightness of mass spectrometry signals? Or brightness of DAPI signal?
24. Line 379: Should read "...saline was injected..."
25. Line 381: Should read "The hypothalamus was collected..."
26. Line 404: "dorsal-ventral"
27. Lines 416-419: For glucose tolerance tests, ad libitum fed mice were used for assessing the effects of inhibitors, while fasted mice were used for assessing the phenotype of cPLA2KD/Sf1 mice. Why were the feeding conditions different among the experiments? Please explain in the manuscript.
28. Fig. 3a: cPLA²KD/Sf1
29. Figs. 3i and 5m: "Suppression" in the label of the y-axis.

Reviewer #2 (Remarks to the Author):

The manuscript from Lee and colleagues sets out to determine how phospholipid metabolism in glucose-sensitive neurons of the ventromedial hypothalamus (VMH) might control glucose homeostasis and influence inflammatory responses. They examine these parameters on a standard diet (SD) and high-fat diet (HFD). Using elegant mass spectrometry imaging (IMS) of the VMH and the arcuate nucleus, they show some impressive differences in phospholipid species in response to glucose injections and HFD. After finding that several AA:PL species are decreased, they then focus on arachidonic acid (AA) and downstream prostaglandins levels. In both hyperglycemia and HFD, prostaglandins, but not the AA precursor are elevated. This is the most interesting and definitive feature of their study. The rest of the story is rather weak and unfocused at this juncture with many "loose ends" regarding both glucose control and inflammation. To follow up on the IMS, they

use a PLA2-inhibitor and Viral Knock-down (KD) of PLA2 (generates AA from PLs) to make the claim that PLA activity (and prostaglandins) are needed for sensing glucose in the dorsomedial VMH (dmVMH). This conclusion was reached as glucose intolerance and hypothalamic neuroinflammation are ameliorated after KD of cPLA2. Neither of these stories is complete. Several key questions remain, such as: how and where AA/prostaglandins produced in the dmVMH act to control glucose homeostasis, and why the arcuate nucleus appears uniquely sensitive to the inflammatory consequences of cPLA2-dependent phospholipid metabolism by the dmVMH. The mass spec analysis of phospholipid levels as well as comprehensive measurements of glucose handling in mice via euglycemic clamp are strengths of this manuscript. The overall topic is of interest to researchers investigating central regulation of metabolism.

Major Comments:

1. Pla2g4a knockdown by shRNA is a critical component of the authors' experimental approach; however, only qPCR is provided as validation, which shows an approximately 50% reduction in transcript levels (Fig S3C). Given that cPLA2 is an enzyme, they must show how much enzymatic activity is changed using a KD approach versus a KO approach. Currently they rely, exclusively on metabolic parameters and cFos staining. Does this KD result in the expected and altered levels phospholipid/prostaglandins?

2. Pla2g4a knockdown appears to block the activation of dmVMH neurons by glucose (Fig 3M,N), suggesting that cPLA2 is required for this response. As HFD increases the biochemical outputs of cPLA2 (Fig 4g); why don't HFD fed mice show activity-dependent cFos expression in response to glucose. This seems paradoxical if cPLA2 in the VMH is required for glucose sensing. This is more worrisome as the VMH is spared from neuroinflammation (Fig 6)?

3. The authors present several lines of evidence suggesting that the dorsomedial VMH subdivision dmVMH mediates the conversion of phospholipid \diamond prostaglandin and regulation of blood glucose: 1) Sf1-Cre, which is used to drive expression of the shRNA knockdown cPLA2 is expressed only in the dmVMH of adult mice (Cheung et al., 2013); and 2) pharmacological inhibition of PLA/COX1/2 block neuronal activity marker expression in the dmVMH. Given these two factors, it is unclear why the authors conclude: "results suggest that cPLA2 in in the VMH has a deteriorative role in the glucose responsiveness of the vVMH/ARC" (lines 236-7).

4. Does Pla2g4a knockdown alter neuronal activation (cFos induction) in response to other metabolic cues such as leptin? Or is it specific to glucose?

5. In Figs 2, 3, 5, and 6 the quantification and images shown are not convincing. Thus, while the subtle differences in cFos, Iba1, and GFAP staining are appreciated, the bar graphs aren't reflected of the images shown and it is worrisome that there is background staining. Presumably, this a consequence of scaling the number of counted cells so that they can be normalized on a per mm² basis (?). Consider changing the normalization to more accurately reflect the data.

6. Assessing inflammatory markers are relevant to the question but there are two major concerns with their analyses. First, sex-differences are well known for HFD and inflammation – why were only male mice used. Relevant to this, Xu's group recently showed that neurons in the VMHvl are glucose sensing (2020). Second, the Sf-1Cre is well known to mark a population of cells just below the ARC in addition to the VMH. Could KD of PLA2 in these cells/neurons account for results shown in Fig 6.

7. The VMH has been shown to be involved in the counter-regulatory response, was this examined and how might their results change in a fasting-refeeding paradigm rather than relying on an IP glucose injection.

Minor Comments:

1. Composition of high-fat diet provided, please ensure that this information is provided.
2. How long after PLA2 knockdown were body and organ weights assessed (Figure S3)?
3. In Figs 3D and 5H, it is unclear what asterisks on graphs refer to (timepoints, groups etc). For all figures, I could not find details on the results of statistical testing beyond a summary of p values. According to the Methods, both Bonferroni and Sidak tests were used for multiple

comparison testing; however, it is not clear which tests were used in which experiment, nor is it clear why the same correction was not used across experiments?

4. Figure 2D, the scale bar is missing.

5. Bar graphs should be changed to show individual data points, especially for physiological endpoints.

Reviewer #3 (Remarks to the Author):

The paper by Ming-Liang Lee et al. explores the interplay between hypothalamic lipid metabolism, glucose metabolism and insulin sensitivity which is essential to expedite the understanding of the development of diabetes and obesity. The authors demonstrate that cPLA2-mediated hypothalamic phospholipid metabolism in mice fed a regular chow diet is essential to control systemic glucose metabolism, and that COX1/2-derived prostaglandins produced from phospholipids activate hypothalamic VMH neurons which renders increased muscle insulin sensitivity. High fat diet in mice increases the hypothalamic cPLA2 activity and renders increased production of downstream prostaglandins, particularly in Sf1 neurons which leads to increased neuroinflammation. The study provides compelling data to support these conclusions, however further improvements are necessary to recommend it for publication.

1. Imaging mass spectrometry experiments were carried out only in negative ion mode, with only a restricted set of phospholipids reported and analysed in the manuscript. Other phospholipids like phosphatidylglycerol, phosphatidylcholine and lysospecies were not considered and discussed in their investigation. The authors should provide a reasoning for this selective analysis and consideration of PLs.

2. Arachidonic acid was not quantified by LC-MS or reported along with prostaglandins in the hypothalamus region. Indeed, AA was quantified in slices by IMS. However, it would be important to understand whether the AA pool is still increased in the hypothalamic region under HFD despite increased production of AA-derived prostaglandins or the cPLA2-mediated AA increase is entirely used for pro-inflammatory lipid synthesis, e.g. prostaglandins. This is also important in view of the fact that AA is also partly used for membrane remodelling, via the AA-CoA, by what has been termed the Lands cycle and the authors should discuss this aspect even if temporal dynamics of AA-containing phospholipids and AA-derived prostaglandins is not evaluated.

3. In the Discussion section, page 15 the statement "Unexpectedly, AA-containing phospholipids decreased because of an increase in hypothalamic cPLA2 activity" should be reconsidered and rephrased, as it is not unexpected that AA-containing phospholipids decrease upon increased cPLA2 activity.

4. The study does not indicate how long is the prostaglandin-induced neuroinflammation lasting and this should be acknowledged in the discussion as a limitation of the study and as a need for perspective investigation.

5. Methods section, Quantification of prostaglandins- please provide detailed description of the LC/MS method used, and/or which specific modifications of the LC/MS method published by Yamada M et al (J. Chromatography B, 2015) were applied.

We thank all reviewers for constructive comments. We have changed our manuscript according to your comments.

REVIEWER COMMENTS

Reviewer #1 (Remarks to the Author):

This study investigated the roles of prostaglandin (PG) production in the ventral hypothalamus in glucose homeostasis by using a variety of techniques including imaging mass spectrometry, c-Fos immunohistochemistry, intra-hypothalamic drug injections to inhibit local production of PGs, and genetic knockdown of cPLA2 mRNA selectively in Sf1-expressing neurons. The findings indicate that products of cPLA2, presumably prostaglandins, function for glucose homeostasis under normal, healthy conditions, while these products exert deteriorating effects on glucose homeostasis under high fat diet (HFD) feeding. The findings are novel and important in understanding the hypothalamic mechanisms of the regulation of glucose metabolism, and would also be relevant to diabetes medication. The experiments presented are overall fair. The authors should take care of many, but relatively minor concerns and corrections:

1. In this study, it is very important to make sure that the AAV injected into the VMH did not diffuse into the ARC and that no cPLA2 was knocked down in the ARC, because the VMH and ARC are close each other. It should be explained in the methods how this was confirmed in every mouse injected with the AAVs. Presenting an overlaid drawing that delineates infected hypothalamic areas in all the mice would be helpful to make sure this point.

Response: First, we have now performed immunohistochemistry to stain alpha-MSH, a marker of POMC neurons that locate the dorsal part of the ARC, in AAV-DIO-GFP injected Sf1-cre mice (Supplemental Fig 5h, i). The data suggests that the cell body stained with GFP was not merged with POMC neurons. Therefore, we believe that the cre recombination did not occur in the ARC and cPLA2 was not knocked down in the ARC.

Second, we added the sentence in the method to explain how we confirm the injection site of the AAV (line 548-550). The change in glucose metabolism in cPLA2KD^{Sf1} mice depends on the accuracy of the injection, since the mice with missed-injection site of the virus has no effects on the glucose and insulin tolerance (Supplemental Fig. 6b-d).

2. HFD feeding increased cPLA2 activity per mg hypothalamic protein (Fig. 4g). As suggested in the discussion, HFD feeding might have promoted phosphorylation of cPLA2 to increase

the catalytic activity, but it is also possible that the expression of cPLA2 was increased. Determining which is the case seems to be important in understanding how HFD stimulates tissue inflammation in the hypothalamus.

Response: We have now measured both phosphorylation of cPLA2 and total cPLA2 by immunoblotting, and added in the Figure 4i-k. The data suggests that the activity of cPLA2 is regulated by phosphorylation of Ser505 at least in part, but not by cPLA2 expression. We also added the discussion in Line 393-395.

3. Hypothalamic contents of several PGs are changed after glucose infusion (Fig. 1i) and after HFD feeding (Fig. 4i). Some PGs were commonly increased, while some others were not. The PGs responsible for glucose homeostasis in healthy mice and for the proinflammatory effect of HFD may be different. It would be interesting to discuss the authors' speculation on the responsible PGs (and responsible PG receptor subtypes, if possible).

Response: I.c.v. injection of PGE2, PGD2 or PGF2 α has been reported to increase hepatic glucose production and promote hyperglycemia (Nonogaki K. et. al. Life Sciences, 1997, 60(11):797-807.). However, roles of each PG in the systemic glucose metabolism and proinflammatory effects of HFD are not clear. We added the discussion in Line 407-421.

4. The glucose infusion in this study dosed the mice with 2 g/kg body weight. What physiological condition was mimicked by this dose? Is this considered physiological mimicking diet-induced hyperglycemia? Or more pathological?

Response: 2 g/kg glucose is a regular concentration for GTT in the research field of metabolism. It's a physiological hyperglycemia. We have now measured blood glucose levels in fasting-refeeding paradigm in both indomethacin-injected mice and RCD-cPLA2^{Sfl} mice (Fig. 2e and Fig. 3c). The data also suggests that the hypothalamic PGs are important to decrease postprandial hyperglycemia.

5. The methods section lacks the information on the procedure of blood sampling and of anesthesia for tissue sampling and decapitation. These are essential information.

Response: During GTT, 2DGTT and ITT, a small drop of blood ($\sim 2 \mu$ l) from the tail vein after poking the tail tip with a needle was placed to the a strip of the handheld glucometer (Nipro Free style, Nipro, Osaka, Japan) to measure the blood glucose level. Mice were sacrificed using CO2 asphyxiation or intravenous infusion of thiopental sodium. We added sentences in the "Methods".

6. Line 51: Should read "Obesity can attenuate the function of these nuclei and promote...",

if this statement is not applicable to all obesity conditions.

We have changed the sentence. (line53)

7. Line 87: Should read “Prostaglandins produced from phospholipids…”

We have changed the sentence. (line91)

8. Line 170: Should read: “…were co-injected into the hypothalamus of wild type mice to knock down…”

We have changed the sentence. (line207)

9. Line 178: How to calculate the rate of disappearance should be explained briefly. It is only referred to ref # 53 in the methods, but it is insufficient for readers to understand the result.

We have added the calculation briefly. (line679-688)

10. Line 220: Should read “…we fed cPLA2KD/Sf1 mice with HFD for 8 wk…”

We have changed the sentence. (line264)

11. Line 223: Should read “Unlike RCD-fed mice (Fig. 3b), knockdown of …”

We have changed the sentence. (line267)

12. Lines 225-226: If this study was not the first to show that HFD-feeding eliminates the responses of VMH or ARC neurons to glucose, please cite the earlier paper.

Response: We have added references which indicate the impaired glucose sensing of VHM or ARC after HFD feeding in line 277-278.

13. Lines 228-229: Should read “…suggesting that cPLA2 knockdown improved neuronal responsiveness to glucose under the HFD-fed condition (Fig. 5d-f).”

We have changed the sentence. (line281)

14. Lines 231-232: Should read “To maintain euglycemia, significantly higher GIR was required in cPLA2KD/Sf1-HFD…”

We have changed the sentence. (line288-289)

15. Lines 231-233: For readers outside the field, it should be briefly explained why GIR was lower in HFD-fed mice (Fig. 5h, black) than RCD-fed mice (Fig. 3d, black).

We have changed the sentence. (line285-288)

16. Line 233: Should read “Unlike RCD-fed mice (Fig. 3f,g), glucose…”

We have changed the sentence. (line290)

17. Lines 237-238: The wording “…attenuates hepatic insulin sensitivity during HFD-induced obesity” sounds problematic for two reasons: 1) the conducted experiments did not examine insulin sensitivity exclusively in the liver, but insulin sensitivity in skeletal muscles, for example, might be altered as well; 2) the HFD-fed mice in this study did not exhibit obesity.

Response: 1) The hyperinsulinemic-euglycemia clamp experiment is one of the best ways to measure insulin sensitivity in the liver. The endogenous glucose production (EGP) is a value to show glucose production (gluconeogenesis and glycogenolysis) from the liver to the circulation, but it also includes gluconeogenesis from the kidney and other tissues. It is known that more than 90% of glucose production is from liver and insulin can suppress hepatic glucose production. Therefore, the suppression of EGP by insulin (Fig 3k, Fig 5m) mainly represents the hepatic insulin sensitivity. Because glucose can be produced by kidney and other tissues via gluconeogenesis, we used EGP to describe it accurately, even though some researchers use hepatic glucose production instead of EGP. But it may make confusions. We added the explanation of EGP in the “Methods”, line 679-688.

2) 8 weeks of HFD was not enough to reach ~50g body weight in our experiments. We stop using “HFD-induced obesity” or “diet-induced obesity” to avoid misunderstanding.

18. Line 261: Should read “in skeletal muscles”.

We have changed the sentence. (line325)

19. Line 265: Is there any evidence that HFD feeding attenuated insulin sensitivity exclusively in the liver?

Response: Same as the comment 17, the suppression of EGP by insulin represents the hepatic insulin sensitivity. % suppression of EGP was 100% in RCD-fed GFP^{Sf1} mice (Fig 3k), while it was 50% in HFD-fed GFP^{Sf1} mice (Fig 5m) in our experiments. The data suggests that a HFD feeding attenuated insulin sensitivity in the liver. The effect of HFD on insulin resistance is mediated by cPLA2 in Sf1 neurons, because cPLA2KD^{Sf1} mice has higher insulin sensitivity in the liver, i.e., suppression of EGP, compared to HFD-fed GFP^{Sf1} mice (Fig 5m).

20. Line 297: Should read “…and are required to…”

We have changed the sentence. (line370)

21. Lines 310-311: What FAs are meant by “all the FAs”?

We have changed the sentence. We added the fatty acids we measured in line 391-392.

22. Line 339: Should read “...has a critical role in inducing...”

We have changed the sentence. (line461)

23. Line 374: brightness of WHAT? Brightness of mass spectrometry signals? Or brightness of DAPI signal?

It was “brightness of mass spectrometry signals”. We have changed the sentence. (line499)

24. Line 379: Should read “...saline was injected...”

We have changed the sentence. (line504)

25. Line 381: Should read “The hypothalamus was collected...”

We have changed the sentence. (line506)

26. Line 404: “dorsal-ventral”

We have changed the sentence. (line538)

27. Lines 416-419: For glucose tolerance tests, ad libitum fed mice were used for assessing the effects of inhibitors, while fasted mice were used for assessing the phenotype of cPLA2KD/Sf1 mice. Why were the feeding conditions different among the experiments? Please explain in the manuscript.

Response: VMH is known to regulate counter regulatory responses (CRR) during hypoglycemia. We first thought that the fasting may affect neuronal activity of glucose inhibited neurons in the VMH to evoke CRR. However, blood glucose levels after fasting or 2DG injection was similar between cPLA2KD^{Sf1} and GFP^{Sf1} mice, suggesting that the cPLA2 in the VMH has minor role in the CRR. Thus, we started to use fasted mice because it can set the feeding condition similar in each animal. We have added the discussion in line 375-381.

28. Fig. 3a: cPLA2KD/Sf1

We have changed the figure.

29. Figs. 3i and 5m: “Suppression” in the label of the y-axis.

We have changed the figures.

Reviewer #2 (Remarks to the Author):

The manuscript from Lee and colleagues sets out to determine how phospholipid metabolism in glucose-sensitive neurons of the ventromedial hypothalamus (VMH) might control glucose homeostasis and influence inflammatory responses. They examine these parameters on a standard diet (SD) and high-fat diet (HFD). Using elegant mass spectrometry imaging (IMS) of the VMH and the arcuate nucleus, they show some impressive differences in phospholipid species in response to glucose injections and HFD. After finding that several AA:PL species are decreased, they then focus on arachidonic acid (AA) and downstream prostaglandins levels. In both hyperglycemia and HFD, prostaglandins, but not the AA precursor are elevated. This is the most interesting and definitive feature of their study. The rest of the story is rather weak and unfocused at this juncture with many “loose ends” regarding both glucose control and inflammation. To follow up on the IMS, they use a PLA2-inhibitor and Viral Knock-down (KD) of PLA2 (generates AA from PLs) to make the claim that PLA activity (and prostaglandins) are needed for sensing glucose in the dorsomedial VMH (dmVMH). This conclusion was reached as glucose intolerance and hypothalamic neuroinflammation are ameliorated after KD of cPLA2. Neither of these stories is complete. Several key questions remain, such as: how and where AA/prostaglandins produced in the dmVMH act to control glucose homeostasis, and why the arcuate nucleus appears uniquely sensitive to the inflammatory consequences of cPLA2-dependent phospholipid metabolism by the dmVMH. The mass spec analysis of phospholipid levels as well as comprehensive measurements of glucose handling in mice via euglycemic clamp are strengths of this manuscript. The overall topic is of interest to researchers investigating central regulation of metabolism.

Major Comments:

1. Pla2g4a knockdown by shRNA is a critical component of the authors' experimental approach; however, only qPCR is provided as validation, which shows an approximately 50% reduction in transcript levels (Fig S3C). Given that cPLA2 is an enzyme, they must show how much enzymatic activity is changed using a KD approach versus a KO approach. Currently they rely, exclusively on metabolic parameters and cFos staining. Does this KD result in the expected and altered levels phospholipid/prostaglandins?

Response: We have measured the cPLA2 activity in RCD-fed mice. However, a glucose injection did not change cPLA2 activity (Fig S5d). Therefore, a measurement of cPLA2 activity is not suitable for the evaluation of shRNA. The reason why the glucose injection did not change cPLA2 activity is that the activity was measured with supernatants of hypothalamic

homogenates. As cPLA2 activity is regulated by cytosolic calcium concentrations, a mixture of tissue homogenate may change physiological intracellular calcium concentration and thus we could not measure physiological cPLA2 activity in vitro (We added this in line 182-186). Therefore, we have measured the change of phospholipids in cPLA2KD^{Sf1} animals to analyze the effect of shRNA.

In the cPLA2KD^{Sf1} mice, a glucose injection did not change the amounts of AA-containing phospholipids in the VMH, but it did decrease them in the ARC (Supplemental Fig. 5e-g). Therefore, the Pla2g4a knockdown by shRNA is effective to inhibit the AA release from the phospholipid by cPLA2.

2. Pla2g4a knockdown appears to block the activation of dmVMH neurons by glucose (Fig 3M,N), suggesting that cPLA2 is required for this response. As HFD increases the biochemical outputs of cPLA2 (Fig 4g); why don't HFD fed mice show activity-dependent cFos expression in response to glucose. This seems paradoxical if cPLA2 in the VMH is required for glucose sensing. This is more worrisome as the VMH is spared from neuroinflammation (Fig 6)?

Response: We appreciate this comment, because it picks up the insufficient point of this paper, which we didn't recognize.

We checked all the data of Iba1 and realized that HFD increased the cell size even in the VMH. We only measured cell number in the previous version. Since the immunofluorescent picture is not suitable to show the size of Iba1 positive cells, we have now stained them with DAB (Fig 6). The data suggests that HFD increases inflammation in both VMH and ARC, and cPLA2KD suppresses them. It is known that the HFD-induced inflammation impairs glucose sensing in the hypothalamus. We have changed the manuscript and we now propose that cPLA2 affects inflammation in the whole VMH and the ARC in HFD-fed mice, and thus HFD attenuates glucose sensing of the dmVMH.

3. The authors present several lines of evidence suggesting that the dorsomedial VMH subdivision dmVMH mediates the conversion of phospholipid \diamond prostaglandin and regulation of blood glucose: 1) Sf1-Cre, which is used to drive expression of the shRNA knockdown cPLA2 is expressed only in the dmVMH of adult mice (Cheung et al., 2013); and 2) pharmacological inhibition of PLA/COX1/2 block neuronal activity marker expression in the dmVMH. Given these two factors, it is unclear why the authors conclude: "results suggest that cPLA2 in in the VMH has a deteriorative role in the glucose responsiveness of the vVMH/ARC" (lines 236-7).

Response: We agree that our previous conclusion was confusing because roles of cPLA2 in

the RCD and HFD are different, i.e., cPLA2 regulates glucose responsiveness of the dmVMH, but not in the vlVMH and ARC, in RCD-fed mice, and cPLA2 impairs glucose responsiveness of the vlVMH and ARC in HFD-fed mice. We have now changed the conclusion according to the new data in Fig 6. Our new conclusion is that prostaglandin production originated from the dmVMH affects microglial activation in the whole VMH and ARC, thus HFD impairs the glucose responsiveness of the whole VMH and ARC. We changed the discussion about the result of cFos in HFD (line 440-448).

4. Does Pla2g4a knockdown alter neuronal activation (cFos induction) in response to other metabolic cues such as leptin? Or is it specific to glucose?

Response: We have now added the data of leptin injected mice. Unfortunately, because all of our Sf1-cre mice were used in the other experiments, we used MAFP i.c.v. to check the role of cPLA2 in leptin-induced cFos induction. The data suggests that the cPLA2 is important for the leptin-induced increase in cFos expression in dmVMH (Fig. S4). We added sentences in the result (line 166-172) and discussion (line 364-366).

5. In Figs 2, 3, 5, and 6 the quantification and images shown are not convincing. Thus, while the subtle differences in cFos, Iba1, and GFAP staining are appreciated, the bar graphs aren't reflected of the images shown and it is worrisome that there is background staining. Presumably, this a consequence of scaling the number of counted cells so that they can be normalized on a per mm² basis (?). Consider changing the normalization to more accurately reflect the data.

Response: We have changed all the representative pictures of cFos, Iba1 and GFAP. We also counted the cell number again and the data is shown as "cFos positive cells/section". We hope that the differences between groups are clear and the new data meets the standard of reviewers.

6. Assessing inflammatory markers are relevant to the question but there are two major concerns with their analyses. First, sex-differences are well known for HFD and inflammation – why were only male mice used. Relevant to this, Xu's group recently showed that neurons in the VMHvl are glucose sensing (2020). Second, the Sf-1Cre is well known to mark a population of cells just below the ARC in addition to the VMH. Could KD of PLA2 in these cells/neurons account for results shown in Fig 6.

Response: First, we have now added the data of female cPLA2KD^{Sf1} mice fed with HFD (Fig. S11 and S12). Our data suggests that the 8 weeks HFD-induced a similar response of hypothalamic inflammation in female mice, and knockdown of cPLA2 in the Sf1 neurons can also ameliorate inflammation and systemic glucose metabolism in female mice (we added the

discussion in line 396-400). We appreciate the reviewer to inform the paper. We added the discussion about glucose sensing in the vVMH (line 359-364). Our cFos data also shows that the vVMH has the character of glucose sensing neurons. However, the cPLA2 has a minor role in the glucose sensing by vVMH in RCD-fed mice, since the icv injection of inhibitors for cPLA2 or COX1/2 did not affect the glucose-induced increase in cFos expression in the vVMH.

Second, we have stained POMC neurons in AAV-DIO-GFP injected Sf1-cre mice to check the localization of Sf1 neurons in the ARC, because POMC neurons exist in the dorsolateral part of the ARC (Fig. S5h, i). The data suggests that the cell body stained with GFP was not merged with POMC neurons. Therefore, we believe that the cre recombination did not occur in the ARC and cPLA2 was not knocked down in the ARC.

In fact, some POMC neurons express Sf1 in the developmental brain (E10.5), but POMC neurons and Sf1 neurons become separated at E12.5 (McNay et al 2006, Molecular Endocrinology.). Therefore, the Sf1-mediated Cre expression may exist only in the VMH in adult mice and the Cre-recombination did not occur in the ARC. If we use cPLA2 floxed mice to make Sf1-Cre-induced knockout mice, cPLA2 would be deleted in some POMC neurons at the developmental stage.

7. The VMH has been shown to be involved in the counter-regulatory response, was this examined and how might their results change in a fasting-refeeding paradigm rather than relying on an IP glucose injection.

Response: We have now measured blood glucose levels after 2-deoxy glucose injection, which promotes glucose deprivation, in RCD-cPLA2KD^{Sf1} mice (Fig 3d). There were no significant differences between GFP^{Sf1} and cPLA2KD^{Sf1} mice. We also measured blood glucose levels in fasting-refeeding paradigm in both indomethacin-injected mice and RCD-cPLA2KD^{Sf1} mice (Fig. 2e and Fig. 3c). The data suggests that the hypothalamic PGs are important to decrease postprandial hyperglycemia.

Minor Comments:

1. Composition of high-fat diet provided, please ensure that this information is provided.

Response: We added the information of high-fat-diet (line475-476). We decided not to write the whole composition of the HFD because too much information makes our paper difficult to read. The information is written in the website of Research Diet Inc. We can add the information anytime if the reviewer asks.

2. How long after PLA2 knockdown were body and organ weights assessed (Figure S3)?

Response: The mice were sacrificed and tissue weight were measured 8 weeks after viral injection. We have added the information in the figure legend.

3. In Figs 3D and 5H, it is unclear what asterisks on graphs refer to (timepoints, groups etc). For all figures, I could not find details on the results of statistical testing beyond a summary of p values. According to the Methods, both Bonferroni and Sidak tests were used for multiple comparison testing; however, it is not clear which tests were used in which experiment, nor is it clear why the same correction was not used across experiments?

Response: We have changed the asterisks on the Fig3f and Fig 5h to compare each data point between groups. We changed the statistic method and we now use only Sidak test in all multiple comparison testing. The manuscript was corrected in line 694.

4. Figure 2D, the scale bar is missing.

Response: The scale bar has been added in Fig. 2f.

5. Bar graphs should be changed to show individual data points, especially for physiological endpoints.

Response: We have change types of all the bar graphs showing individual data points.

Reviewer #3 (Remarks to the Author):

The paper by Ming-Liang Lee et al. explores the interplay between hypothalamic lipid metabolism, glucose metabolism and insulin sensitivity which is essential to expedite the understanding of the development of diabetes and obesity. The authors demonstrate that cPLA2- mediated hypothalamic phospholipid metabolism in mice fed a regular chow diet is essential to control systemic glucose metabolism, and that COX1/2-derived prostaglandins produced from phospholipids activate hypothalamic VMH neurons which renders increased muscle insulin sensitivity. High fat diet in mice increases the hypothalamic cPLA2 activity and renders increased production of downstream prostaglandins, particularly in Sf1 neurons which leads to increased neuroinflammation. The study provides compelling data to support these conclusions, however further improvements are necessary to recommended it for publication.

1. Imaging mass spectrometry experiments were carried out only in negative ion mode, with only a restricted set of phospholipids were reported and analysed in the manuscript. Other

phospholipids like phosphatidylglycerol, phosphatidylcholine and lysospecies were not considered and discussed in their investigation. The authors should provide a reasoning for this selective analysis and consideration of PLs.

Response: We have included the data of lysospecies, such as lysophosphatidyl-inositol, -ethanolamine and -serine in the hypothalamus. Glucose injection and HFD feeding both tended to increase these lysospecies compared to RCD saline injected group (Fig. S1 and Fig. S8). The increase in lysospecies was not obvious, suggesting that the lysospecies may be used in the recycling or as an agonist of G-protein coupled receptor 55 (GPR55), which is known to regulate metabolism (Meadows A, et al., 2016, *Int J Obes*). We added the text in the discussion (line 421-430).

We have also tried to measure phosphatidylglycerol and phosphatidylcholine with positive ion mode. However, the MS/MS analysis was not successful so far. We would like to try this experiment in future.

2. Arachidonic acid was not quantified by LC-MS or reported along with prostaglandines in the hypothalamus region. Indeed, AA was quantified in slices by IMS. However, it would be important to understand whether the AA pool is still increased in the hypothalamic region under HFD despite increased production of AA-derived prostaglandins or the cPLA2 mediated AA increase is entirely used for pro-inflammatory lipid synthesis, e.g. prostaglandines. This is also important in view of the fact that AA is also partly used for membrane remodelling, via the AA-CoA, by what has been termed the Land's cycle and the authors should discuss this aspect even if temporal dynamic of AA-containing phospholipids and AA-derived prostaglandines is not evaluated.

Response: We have added the data of AA measured by LC-MS (Fig 4m). The amount of AA tends to be higher in the HFD-fed mice, suggesting that the AA pool in the hypothalamus is higher in the HFD-fed mice compared to RCD-fed mice. AA may be partly used for membrane remodeling or the other purpose. We have added the discussion (line422-426).

3. In the Discussion section, page 15 the statement " Unexpectedly, AA-containing phospholipids decreased because of an increase in hypothalamic cPLA2 activity" should be reconsidered and rephrased, as it is not unexpected that AA-containing phospholipids decrease upon increased cPLA2 activity

Response: We regret our mistake. We have changed the word. (line 392)

4. The study does not indicate how long is the prostaglandin- induced neuroinflammation lasting and this should be acknowledged in the discussion as a limitation of the study and as a

need for perspective investigation.

Response: We appreciate the insightful comment. We have added the discussion (line 403-406).

5.Methods section, Quantification of prostaglandins- please provide detailed description of the LC/MRM method used, and/or which specific modifications of the LC/MS method published by Yamada M et al (J. Chromatography B, 2015) were applied.

Response: We have add the detailed description in Method section (line 519-530) and Table S3.

Reviewer comments, second round:

Reviewer #1 (Remarks to the Author):

After the revision, the manuscript has been greatly improved, but there are some minor corrections in the text, which the authors would like to take care of. The manuscript should be carefully checked before publication.

1. Line 82: "Other PUFAs, such as oleic acid (OA), modulate activities..."
2. Line 185: "...concentration was not conserved."
3. Line 194: In the image (Fig. 5h), POMC neurons appear to be distributed in the main part of the ARC, but fewer in the "dorsal part".
4. Line 201: "...the virus had no effects..."
5. Lines 240 and 311: "...both VMH and ARC..."
6. Line 254: "...AA tended to be higher..."
7. Line 270: "...observed in female mice."
8. Line 278: "...reported in previous studies..."
9. Line 302: "astrocyteS"
10. Line 370: "Leptin receptors are located in the dmVMH and required to..."
11. Line 378: "blood glucose levels after fasting or 2DG injection were similar..."
12. Line 609: "Sections were then incubated..."
13. Line 636: "...sections containing VMH and ARC were..."

Reviewer #2 (Remarks to the Author):

The authors have addressed much of this reviewer's concerns and have modified the manuscript accordingly to improve its overall impact and rigor. Among these changes, the authors have expanded their analysis of cPLA2 involvement in the HFD response to include female mice and have revised the text to more accurately reflect the data.

Minor comments/suggestions to improve the readability of the paper:

1. Using "gender" to refer to male and female mice (line 399) is inaccurate, consider replacing it with the word "sex" or "biological sex".
2. The statistical differences in VMH microglia size (Figures 6 and S12) seem to be overestimated as the number of N's is based on the total number of microglia scored (~30-40) rather than the number of mice (N=3).
3. Please provide a brief statement in the methods regarding how astrocyte and microglia cell numbers and sizes were quantified
4. There are several typos throughout, including:
line 269: "The similar results..."
line 270: "The body weight was..."
line 386: "that microglia was..."

line 321: "...homeostasis is ill-defined"
line 423: "...as known as..."
line 635: "...started to fed..."

5. The graphical abstract needs to be improved and perhaps simplified to make only the most important points of the study. Along those lines, for those not familiar with lipid metabolism, it would help to have the major species discussed clearly annotated in better graphics in Figure 1I and Figure 24I.

6. The distinction between the ARC and VMH continues to remain puzzling. Right now it is clear that KD of PLA2 using the SF-1 Cre affects aspects of both the ARC and VMH. But excluding the effects of lipid synthesis in the ARC at this juncture, (in the absence of carrying out the equivalent KD in the ARC) might lead to a premature and inaccurate conclusion. It is suggested that the text be modified to make this point.

Reviewer #3 (Remarks to the Author):

The authors have addressed the issues raised including the requested experimental data and the corresponding discussion in the manuscript.

I would recommend the publication of the manuscript provided that the following minor revision is carried out:

1. In the Supplemental Material, the Table sheet corresponding to Figure S9 is empty. The data need to be added in the excel sheet.

2. The authors need to include a statement/explanation in the manuscript concerning the lack of PC and PG data indicating that the imaging MS/MS data for these two molecular categories or generally for positive ion mode were not unambiguous or conclusive to allow a reliable data assignment. Please provide information in the manuscript whether the IMS data in positive ion mode were collected from the same tissue slices as for negative ion mode or from adjacent or subsequent tissue slices. These information are important for the reader to understand the study design, as well as the limitations of the study.

We thank all reviewers for constructive comments. We have changed our manuscript according to your comments.

REVIEWERS' COMMENTS

Reviewer #1 (Remarks to the Author):

After the revision, the manuscript has been greatly improved, but there are some minor corrections in the text, which the authors would like to take care of. The manuscript should be carefully checked before publication.

1. Line 82: "Other PUFAs, such as oleic acid (OA), modulate activities..." Line 82
2. Line 185: "...concentration was not conserved." Line 185
3. Line 194: In the image (Fig. 5h), POMC neurons appear to be distributed in the main part of the ARC, but fewer in the "dorsal part". Line 194-195
4. Line 201: "...the virus had no effects..." Line 202
5. Lines 240 and 311: "...both VMH and ARC..." Line 241 and 312-313
6. Line 254: "...AA tended to be higher..." Line 255
7. Line 270: "...observed in female mice." Line 271
8. Line 278: "...reported in previous studies..." Line 279
9. Line 302: "astrocyteS" Line 303
10. Line 370: "Leptin receptors are located in the dmVMH and required to..." Line 371
11. Line 378: "blood glucose levels after fasting or 2DG injection were similar..." Line 379
12. Line 609: "Sections were then incubated..." Line 618
13. Line 636: "...sections containing VMH and ARC were..." Line 649

Response: We appreciate your explicit corrections. We have changed these sentences.

Reviewer #2 (Remarks to the Author):

The authors have addressed much of this reviewer's concerns and have modified the manuscript accordingly to improve its overall impact and rigor. Among these changes, the authors have expanded their analysis of cPLA2 involvement in the HFD response to include female mice and have revised the text to more accurately reflect the data.

Minor comments/suggestions to improve the readability of the paper:

1. Using “gender” to refer to male and female mice (line 399) is inaccurate, consider replacing it with the word “sex” or “biological sex”.

Response: We appreciate your accurate English. We have changed these sentences in Line 400.

2. The statistical differences in VMH microglia size (Figures 6 and S12) seem to be overestimated as the number of N's is based on the total number of microglia scored (~30-40) rather than the number of mice (N=3).

Response: We are now showing the figures as n=3. It is statistically more accurate now.

3. Please provide a brief statement in the methods regarding how astrocyte and microglia cell numbers and sizes were quantified

Response: We have added how we measured them in the Method (line 633-637).

4. There are several typos throughout, including:

line 269: “The similar results...” Line 270

line 270: "The body weight was..." Line 271

line 386: “that microglia was...” Line 387

line 321: “...homeostasis is ill-defined” Line 322

line 423: “...as known as...” Line 424

line 635: “...started to fed...” Line 648

Response: We appreciate these corrections. We have changed them.

5. The graphical abstract needs to be improved and perhaps simplified to make only the most important points of the study. Along those lines, for those not familiar with lipid metabolism, it would help to have the major species discussed clearly annotated in better graphics in Figure 1I and Figure 24I.

Response: We have changed the graphical abstract. We hope that it is simple enough and shows the most important points of the study.

We agree that Fig 1I and 4I were not easy for researchers outside of the field. We have arranged the position of words, annotated major species of prostaglandin that were changed in this study and added a sentence in the figure legends (line 924 and 1024).

6. The distinction between the ARC and VMH continues to remain puzzling. Right now

it is clear that KD of PLA2 using the SF-1 Cre affects aspects of both the ARC and VMH. But excluding the effects of lipid synthesis in the ARC at this juncture, (in the absence of carrying out the equivalent KD in the ARC) might lead to a premature and inaccurate conclusion. It is suggested that the text be modified to make this point.

Response: This is very true. We shouldn't exclude the importance of prostaglandin production and lipid synthesis in the ARC. We have added sentences in the discussion (line 449-454).

Reviewer #3 (Remarks to the Author):

The authors have addressed the issues raised including the requested experimental data and the corresponding discussion in the manuscript.

I would recommend the publication of the manuscript provided that the following minor revision is carried out:

1. In the Supplemental Material, the Table sheet corresponding to Figure S9 is empty. The data need to be added in the excel sheet.

Response: We regret the mistake. We have added the data of Figure S9 in the excel sheet.

2. The authors need to include a statement/explanation in the manuscript concerning the lack of PC and PG data indicating that the imaging MS/MS data for these two molecular categories or generally for positive ion mode were not unambiguous or conclusive to allow a reliable data assignment. Please provide information in the manuscript whether the IMS data in positive ion mode were collected from the same tissue slices as for negative ion mode or from adjacent or subsequent tissue slices. These information are important for the reader to understand the study design, as well as the limitations of the study.

Response: Sorry for our insufficient revision. We have added sentences about the measurement of positive ion mode in the "Methods". (line 506-509)